# Compositional Generalization from First Principles

**Thaddäus Wiedemer**[1,2,3*]    **Prasanna Mayilvahanan**[1,2,3*]

**Matthias Bethge**[1,2†]    **Wieland Brendel**[2,3†]

[1]University of Tübingen    [2]Tübingen AI Center
[3]Max-Planck-Institute for Intelligent Systems, Tübingen

`{thaddaeus.wiedemer, prasanna.mayilvahanan}@uni-tuebingen.de`

## Abstract

Leveraging the compositional nature of our world to expedite learning and facilitate generalization is a hallmark of human perception. In machine learning, on the other hand, achieving compositional generalization has proven to be an elusive goal, even for models with explicit compositional priors. To get a better handle on compositional generalization, we here approach it from the bottom up: Inspired by identifiable representation learning, we investigate compositionality as a property of the data-generating process rather than the data itself. This reformulation enables us to derive mild conditions on only the support of the training distribution and the model architecture, which are sufficient for compositional generalization. We further demonstrate how our theoretical framework applies to real-world scenarios and validate our findings empirically. Our results set the stage for a principled theoretical study of compositional generalization.

## 1 Introduction

*Systematic compositionality* [1] is the remarkable ability to utilize a finite set of known components to understand and generate a vast array of novel combinations. This ability, referred to by Chomsky [2] as the "*infinite use of finite means*", is a distinguishing feature of human cognition, enabling us to adapt to diverse situations and learn from varied experiences.

It's been a long-standing idea to leverage the compositional nature of the world for learning. In object-centric learning, models learn to isolate representations of individual objects as building blocks for complex scenes. In disentanglement, models aim to infer factors of variation that capture compositional and interpretable aspects of their inputs, for example hair color, skin color, and gender for facial data. So far, however, there is little evidence that these methods deliver substantially increased learning efficacy or generalization capabilities (Schott et al. [3], Montero et al. [4]). Across domains and modalities, machine learning models still largely fail to capture and utilize the compositional nature of the training data (Lake and Baroni [5], Loula et al. [6], Keysers et al. [7]).

To exemplify this failure, consider a model trained on a data set with images of two sprites with varying position, size, shape, and color overlaid on a black canvas. Given the latent factors, a simple multi-layer neural network can easily learn to reconstruct images containing *compositions* of these sprites that were covered by the training set. However, reconstruction fails for novel compositions— even if the individual *components* have been observed before (see Figure 1). Failure to generalize to

---

*Equal contribution    †Equal supervision

Code available at https://github.com/brendel-group/compositional-ood-generalization

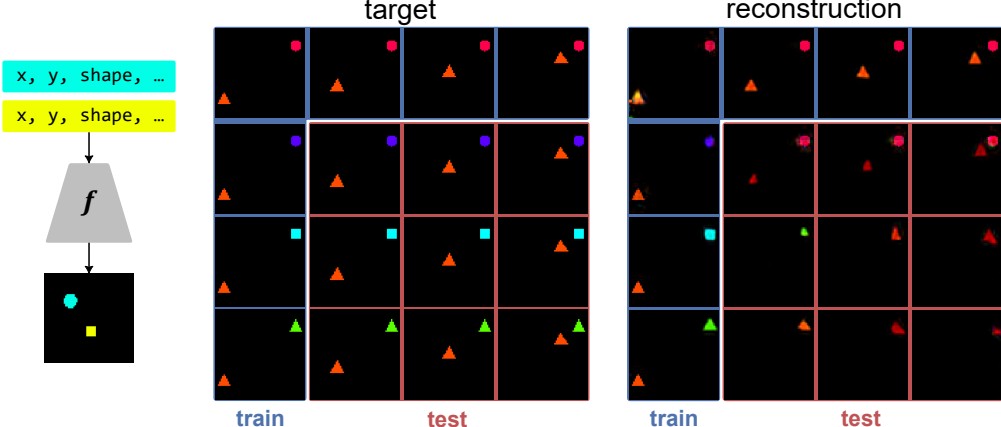

Figure 1: **Compositional generalization fails even in regression settings**. **Left**: We train a model $f$ to reconstruct images containing two sprites given their latent representation (`x`, `y`, `shape`, `size`, `color`). **Center**: In the **training set** (top row and left column), one sprite is fixed in its base configuration (orange triangle or red circle), while the other is varied randomly (in this example, sprite 1 varies in position, sprite 2 in shape and color). As a result, each sample in the **test set** (lower right block) can be expressed as a novel *composition* of known *components*. **Right**: While the model is able to fit the **training data**, it fails to *generalize compositionally* to the **test data.**

unseen data in even this simplistic regression setting demonstrates that *compositional generalization* does not automatically emerge simply because the data is of a compositional nature.

We therefore take a step back to formally study compositionality and understand what conditions need to be fulfilled for compositional generalization to occur. To this end, we take inspiration from identifiable representation learning and define a broad class of data generating processes that are compositional and for which we can prove that inference models can generalize to novel compositions that have not been part of the training set. More precisely, our contributions are as follows:

- We specify *compositional data-generating processes* both in terms of their function class and latent distributions (Sections 3.1 and 3.2) such that they cover a wide range of assumptions made by existing compositional methods.
- We prove a set of sufficient conditions under which models trained on the data are able to generalize compositionally (Section 3.3).
- We validate our theory in a range of synthetic experiments and perform several ablation studies that relate our findings to empirical methods (Section 4).

## 2   Related Work

**Representation learning**   *Disentanglement* and *identifiable representation learning* aim to learn succinct representations that both factorize the data space efficiently and are robust towards distributional changes [8–10]. However, the expectation that more compositional representations lead to better out-of-distribution (OOD) generalization has not been met, as demonstrated by Schott et al. [3] and Montero et al. [11]. Although our work does not directly address generalization issues in identifiable representation learning, our setup is directly inspired by it, and we examine data-generating processes similar to [12–14].

**Empirical Approaches**   Many empirical methods use compositional priors and claim improved compositional generalization. The problem has been studied especially closely in language [15–17], but it remains far from being solved [5–7]. Object-centric learning is another domain in which compositionality plays a major role, and many approaches explicitly model the composition of scenes from object-"slots" [18–22]. The slot approach is also common in vector-symbolic architectures like [23] and [24]. For most of these works, however, compositional generalization

is not a focal point, and their actual generalization capability remains to be studied. There are also some architectures like transformers [25], graph neural networks [26], bilinear models [27], or complex-valued autoencoders [28] that have been claimed to exhibit some degree of compositional generalization, but again, principled analysis of their generalization ability is lacking. Our framework can guide the systematic evaluation of these methods. While we use the visual domain as an example throughout this work, our contributions are not tied to any specific data domain or modality.

**Theoretical approaches to OOD generalization**    The OOD generalization problem for non-linear models where train and test distributions differ in their densities, but not their supports, has been studied extensively, most prominently by Ben-David and Urner [29] and Sugiyama et al. [30]. We refer the reader to Shen et al. [31] for a comprehensive overview. In contrast, compositional generalization requires generalizing to a distribution with different, possibly non-overlapping support. This problem is more challenging and remains unsolved. Ben-David et al. [32] were able to show that models can generalize between distributions with a very specific relation, but it is unclear what realistic distributions fit their constraints. Netanyahu et al. [33] also study *out-of-support* problems theoretically but touch on compositional generalization only as a workaround for general extrapolation. Recently, Dong and Ma [34] took a first step towards a more applicable theory of compositional generalization to unseen domains, but their results still rely on specific distributions, and they do not consider functions with arbitrary (nonlinear) compositions or multi-variate outputs. In contrast, our framework is independent of the exact distributions used for training and testing, and our assumptions on the compositional nature of the data allow us to prove generalization in a much broader setting.

## 3    A framework for compositional generalization

**Notation**    $[N]$ denotes the set of natural numbers $\{1, 2, ..., N\}$. Vector-valued variables (e.g., $\boldsymbol{x}$) and functions (e.g., $\boldsymbol{f}$) are written in bold. **Id** denotes the (vector-valued) identity function. We write the support of a distribution $P$ as $\operatorname{supp} P$. To express that two functions $\boldsymbol{f}, \boldsymbol{g}$ are equal for all points in the support of distribution $P$, i.e., $\boldsymbol{f}(\boldsymbol{x}) = \boldsymbol{g}(\boldsymbol{x})\ \forall \boldsymbol{x} \in \operatorname{supp} P$, we write $\boldsymbol{f} \equiv_P \boldsymbol{g}$. Finally, $\frac{\partial \boldsymbol{f}}{\partial \boldsymbol{x}}$ denotes the total derivative of a vector-valued function $\boldsymbol{f}$ by all its inputs $\boldsymbol{x}$, corresponding to the Jacobian matrix with entries $\frac{\partial f_i}{\partial x_j}$.

### 3.1    Compositionality

Colloquially, the term "*compositional data*" implies that the data can be broken down into discrete, identifiable components that collectively form the whole. For instance, in natural images, these components might be objects, while in music, they might be individual instruments. As a running illustrative example, we will refer to a simple dataset similar to multi-dSprites [21], as shown in Figure 1. Each sample in this dataset is a composition of two basic sprites, each with a random position, shape, size, and color, size.

Drawing inspiration from identifiable representation learning, we define compositionality mathematically as a property of the data-generating process. In our example, the samples are generated by a simple rendering engine that initially renders each sprite individually on separate canvases. These canvases are then overlaid to produce a single image featuring two sprites. More specifically, the rendering engine uses the (latent) properties of sprite one, $\boldsymbol{z}_1 = (z_{1,\mathrm{x}}, z_{1,\mathrm{y}}, z_{1,\mathrm{shape}}, z_{1,\mathrm{size}}, z_{1,\mathrm{color}})$, to produce an image $\tilde{\boldsymbol{x}}_1$ of the first sprite. The same process is repeated with the properties of sprite two, $\boldsymbol{z}_2 = (z_{2,\mathrm{x}}, z_{2,\mathrm{y}}, z_{2,\mathrm{shape}}, z_{2,\mathrm{size}}, z_{2,\mathrm{color}})$, to create an image $\tilde{\boldsymbol{x}}_2$ of the second sprite. Lastly, the engine combines $\tilde{\boldsymbol{x}}_1$ and $\tilde{\boldsymbol{x}}_2$ to create the final overlaid rendering $\boldsymbol{x}$ of both sprites. Figure 2 demonstrates this process.

In this scenario, the individual sprite renderers carry out the bulk of the work. In contrast, the composition of the two intermediate sprite images $\tilde{\boldsymbol{x}}_1, \tilde{\boldsymbol{x}}_2$ can be formulated as a simple pixel-wise operation (see Appendix B.1 for more details). The rendering processes for each sprite are independent: adjusting the properties of one sprite will not influence the intermediate image of the other, and vice versa.

We posit that this two-step generative procedure—the (intricate) generation of individual components and their (simple) composition into a single output—is a key characteristic of a broad class of

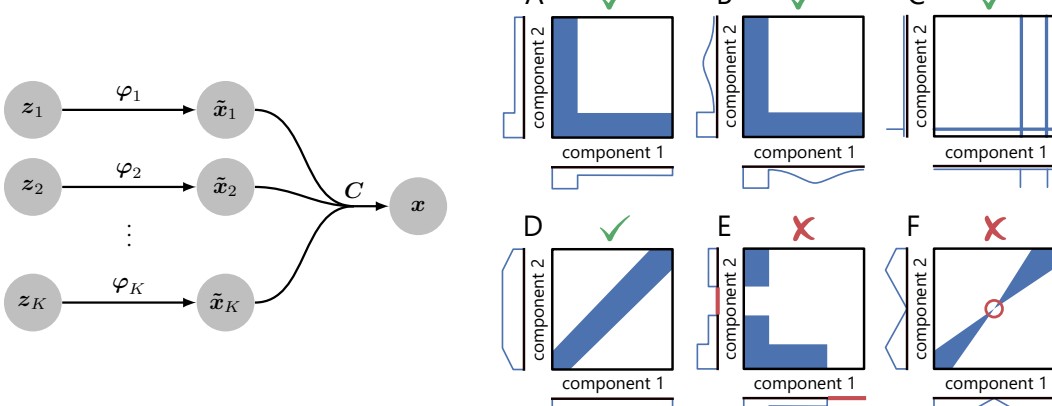

Figure 2: **Compositional representation of a function** (Definition 1). The *component functions* $\varphi_k$ map each *component latent* $z_k$ to an intermediate component representation $\tilde{x}_k$. The *composition function* $C$ composes all component representations into a final data point $x$.

Figure 3: **Compositional support** (Definition 2). **A-D**: Distribution $P$ (blue) has *compositional support* w.r.t. to the entire latent space if it has full support over the marginals. **E**: Gaps in the support require the model to interpolate/extrapolate rather than generalize compositionally. **F**: The support of the joint needs to be in an open set.

compositional problems. If we know the composition function, then understanding the basic elements (for example, the individual sprites) is enough to grasp all possible combinations of sprites in the dataset. We can thus represent any latent variable model $f : \mathcal{Z} \to \mathcal{X}$, which maps a latent vector $z \in \mathcal{Z}$ to a sample $x$ in the observation space $\mathcal{X}$, as a two-step generative process.

**Definition 1** (Compositional representation). $\{C, \varphi_1, \ldots, \varphi_K, \mathcal{Z}_1, \ldots, \mathcal{Z}_K, \tilde{\mathcal{X}}_1, \ldots, \tilde{\mathcal{X}}_K\}$ is a *compositional representation* of function $f$ if

$$\forall z \in \mathcal{Z} \quad f(z) = C\big(\varphi_1(z_1), ..., \varphi_K(z_K)\big) \quad \text{and} \quad \mathcal{Z} = \mathcal{Z}_1 \times \cdots \times \mathcal{Z}_K, \tag{1}$$

where $z_i$ denotes the canonical projection of $z$ onto $\mathcal{Z}_i$. We refer to $\varphi_k : \mathcal{Z}_k \to \tilde{\mathcal{X}}_k$ as the *component functions*, to $\tilde{\mathcal{X}}_1, \ldots, \tilde{\mathcal{X}}_K$ as the (hidden) *component spaces*, and to $C : \tilde{\mathcal{X}}_1 \times \cdots \times \tilde{\mathcal{X}}_K \to \mathcal{X}$ as the *composition function*.

Note that in its most general form, we do not require the component functions to be identical or to map to the same component space. The compositional representation of a function $f$ is also not unique. For instance, any $f$ possesses a trivial compositional representation given by $\{f, \mathbf{Id}, \ldots, \mathbf{Id}\}$ (for the sake of clarity, we will omit the explicit mention of the latent factorization and component spaces henceforth). We will later establish conditions that must be met by at least one compositional representation of $f$.

Our definition of compositionality naturally aligns with various methods in the fields of identifiability, disentanglement, or object-centric learning. In the decoder of SlotAttention [18], for example, each component function is a spatial broadcast decoder followed by a CNN, and the composition function is implemented as alpha compositing. Frady et al. [24] model the component functions as element-wise multiplication of high-dimensional latent codes, which are then composed through a straightforward sum. A similar approach is chosen by Vankov and Bowers [23], except that interactions between components are modeled using matrix multiplication.

## 3.2 Compositional Generalization

The model in Figure 1 was trained supervisedly, i.e., it was trained to reconstruct samples $x$ given the ground-truth latent factors $(z_1, z_2)$ for each sprite (see Section 4 for more details). We denote this model as $\hat{f}$, indicating that it is meant to replicate the ground-truth generating process $f$ of the data. The model $\hat{f}$ indeed learned to fit $f$ almost perfectly on the training distribution $P$, but failed to do so on the test distribution $Q$.

This failure is surprising because the test samples only contain sprites already encountered during training. The novelty lies solely in the combination of these sprites. We would expect any model that comprehends the compositional nature of the dataset to readily generalize to these test samples.

This compositional aspect of the generalization problem manifests itself in the structure of the training and test distribution. In our running example, the model was trained on samples from a distribution $P$ that contained all possible sprites in each slot but only in combination with one base sprite in the other slot (illustrated in Figure 3A). More formally, the support of $P$ can be written as

$$\text{supp } P = \left\{ (z_1 \in \mathcal{Z}_1, z_2 \in \mathcal{Z}_2) | z_1 = z_1^0 \vee z_2 = z_2^0 \right\}, \tag{2}$$

where $z_k^0$ denotes the base configuration of a sprite (e.g., the orange triangle and red square in the samples shown in Figure 1).

The test distribution $Q$ is a uniform distribution over the full product space $\mathcal{Z}_1 \times \mathcal{Z}_2$, i.e., it contains all possible sprite combinations. More generally, we say that a generalization problem is compositional if the test distribution contains only components that have been present in the training distribution, see Figure 3. This notion can be formalized as follows based on the support of the marginal distributions:

**Definition 2** (Compositional support). Given two arbitrary distribution $P, Q$ over latents $z = (z_1, ..., z_K) \in \mathcal{Z} = \mathcal{Z}_1 \times \cdots \times \mathcal{Z}_K$, $P$ has *compositional support* w.r.t. $Q$ if the support over all marginals $P_{z_k}, Q_{z_k}$ is the same:

$$\text{supp } P_{z_k} = \text{supp } Q_{z_k} \subseteq \mathcal{Z}_k \quad \forall k \in [K]. \tag{3}$$

Clearly, *compositional generalization* requires compositional support. If regions of the test latent space exist for which a component is not observed, as in Figure 3E, we can examine a model's generalization capability, but the problem is not compositional. Depending on whether the gap in the support is in the middle of a latent's domain or towards either end, the generalization problem becomes an *interpolation* or *extrapolation* problem instead, which are not the focus of this work.

## 3.3 Sufficient conditions for compositional generalization

With the above setup, we can now begin to examine under what conditions compositional generalization can be guaranteed to occur.

To make this question precise, let us assume for the moment that sprites don't occlude each other but that they are just summed up in pixel space. Then the compositional representation of the generative process is simply $\{\sum(\cdot), \varphi_1, \varphi_2\}$, i.e.

$$\boldsymbol{f}(z) = \varphi_1(z_1) + \varphi_2(z_2). \tag{4}$$

The question becomes: Given training samples $(z, x)$ from $P$, can we train a model $\hat{f}$ that fitting this generative process $f$ on $P$ also guarantees fitting it on $Q$? That is, we are looking for conditions such that the model *generalizes* from $P$ to $Q$:

$$\boldsymbol{f} \underset{P}{\equiv} \hat{\boldsymbol{f}} \implies \boldsymbol{f} \underset{Q}{\equiv} \hat{\boldsymbol{f}}. \tag{5}$$

We assume that $C$ is known, so in order to generalize, we must be able to reconstruct the individual component functions $\varphi_k$. For the simple case from equation 4, we can fully reconstruct the component functions as follows. First, we note that if $\text{supp } P$ is in an open set, we can locally reconstruct the hidden Jacobian of $\varphi_k$ from the observable Jacobian of $f$ as

$$\frac{\partial \boldsymbol{f}}{\partial z_k}(z) = \frac{\partial \varphi_k}{\partial z_k}(z_k). \tag{6}$$

Since the training distribution contains all possible component configurations $z_k$, we can reconstruct the Jacobian of $\varphi_k$ in every point $z_k$. Then we know everything about $\varphi_k$ up to a global offset (which can be removed if there exists a known initial point for integration).

Our goal is to extend this approach to a maximally large set of composition functions $C$. Our reasoning is straightforward if $C$ is the identity, but what if we have occlusions or other nonlinear interactions between slots? What are general conditions on $C$ and the support of the training distribution $P$ such that we can still reconstruct the individual component functions and thus generalize compositionally?

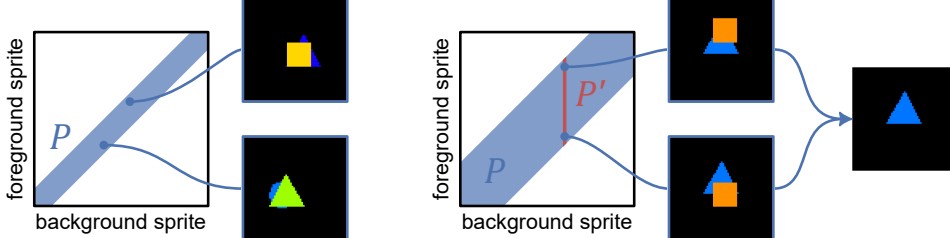

Figure 4: **Sufficient support condition** (Definition 3). For a (compositional) diagonal support, all samples will contain sprites with similar positions, leading to heavy occlusions and making reconstruction of the background sprite impossible (left). Reconstruction of the background sprite is only possible if the support is chosen broad enough, such that the subset of points sharing the same background sprite $P'$ contains samples with sufficient variance in the foreground sample. Specifically, each pixel of the background sprite must be observable at least once (right).

Let us now consider the sprites example with occlusions, where $\varphi_1$ renders the background sprite that is occasionally occluded by the foreground sprite rendered by $\varphi_2$. Let us also assume that the support of $P$ is basically a thin region around the diagonal; see Figure 4 (left). In this case, the two sprites are always relatively similar, leading to large overlaps for practically all samples of the training set. It is impossible to reconstruct the full Jacobian of the occluded sprite from a single sample. Instead, we need a set of samples for which the background sprite is the same while the foreground sprite is in different positions; see Figure 4 (right). With sufficient samples of this kind, we can observe all pixels of the background sprite at least once. Then reconstruction of the Jacobian of $\varphi_1$ is possible again.

This line of thought brings us to a more general condition on the data-generating process: The composition function $C$ and the support of the training set must be chosen such that the full Jacobian can be reconstructed for each component function and for all component latents. In other words, for each configuration of a given component, $P$ must be sufficiently large so that it is possible to track how each dimension of the output depends on each dimension of the component representation. We formally define the concept of *sufficient support* below. Note that whether the support of $P$ is sufficient or not depends on the choice of composition function $C$; see Appendix C for examples.

**Definition 3** (Sufficient support). A distribution $P$ over latents $\boldsymbol{z} = (\boldsymbol{z}_1, ..., \boldsymbol{z}_K) \in \mathcal{Z}$, has *sufficient support* w.r.t. a compositional representation of a function $\boldsymbol{f}$, if $\operatorname{supp} P$ is in an open set and for any latent value $\boldsymbol{z}_k^*$, there exists a finite set of points $P'(\boldsymbol{z}, k) \subseteq \{\boldsymbol{p} \in \operatorname{supp} P | \boldsymbol{p}_k = \boldsymbol{z}_k\}$ for which the sum of total derivatives of $C$ has full rank. That is,

$$\operatorname{rank} \sum_{\boldsymbol{p} \in P'(\boldsymbol{z}, k)} \frac{\partial \boldsymbol{C}}{\partial \boldsymbol{\varphi}_k}\big(\boldsymbol{\varphi}(\boldsymbol{p})\big) = M, \tag{7}$$

where $M$ is the dimension of the component space $\tilde{\mathcal{X}}_k \subseteq \mathbb{R}^M$.

We are now ready to state our main theorem, namely that if $\boldsymbol{f}, \hat{\boldsymbol{f}}$ share the same composition function and if $P$ has compositional and sufficient support, then the model $\hat{\boldsymbol{f}}$ generalizes to $Q$ if it matches the ground-truth data-generating process $\boldsymbol{f}$ on $P$.

**Theorem 1.** *Let $P, Q$ be arbitrary distributions over latents $\boldsymbol{z} = (\boldsymbol{z}_1, ..., \boldsymbol{z}_K) \in \mathcal{Z}$. Let $\boldsymbol{f}, \hat{\boldsymbol{f}}$ be functions with* compositional representations *in the sense of definition 1 that share $\{\boldsymbol{C}, \mathcal{Z}_1, ..., \mathcal{Z}_K\}$, but use arbitrary $\{\boldsymbol{\varphi}_1, ..., \boldsymbol{\varphi}_K, \tilde{\mathcal{X}}_1, ..., \tilde{\mathcal{X}}_K\}, \{\hat{\boldsymbol{\varphi}}_1, ..., \hat{\boldsymbol{\varphi}}_K, \hat{\mathcal{X}}_1, ..., \hat{\mathcal{X}}_K\}$.*

*Assume the following assumptions hold:*

*(A1) $\boldsymbol{C}, \boldsymbol{\varphi}_k, \hat{\boldsymbol{\varphi}}_k$ are differentiable, $\boldsymbol{C}$ is Lipschitz in $\boldsymbol{\varphi}$, and $\boldsymbol{\varphi}$ is continuous in $\boldsymbol{z}$.*

*(A2) $P$ has compositional support w.r.t. $Q$ in the sense of definition 2.*

*(A3) $P$ has sufficient support w.r.t. $\boldsymbol{f}$ in the sense of definition 3.*

*(A4) There exists an initial point $\boldsymbol{p}^0 \in \operatorname{supp} P$ such that $\boldsymbol{\varphi}(\boldsymbol{p}^0) = \hat{\boldsymbol{\varphi}}(\boldsymbol{p}^0)$.*

*Then $\hat{\boldsymbol{f}}$ generalizes to $Q$, i.e. $\boldsymbol{f} \underset{P}{\equiv} \hat{\boldsymbol{f}} \implies \boldsymbol{f} \underset{Q}{\equiv} \hat{\boldsymbol{f}}$.*

The proof follows roughly the intuition we developed above in that we show that the Jacobians of the component functions can be reconstructed everywhere. Bear in mind that this is simply a construction for the proof: The theorem holds whenever $\hat{f}$ fits the output of $f$ on the training distribution $P$, which we can achieve with standard supervised training and without access to the ground-truth Jacobians. It should also be emphasized that since the compositional representation is not unique, the theorem holds if there exists at least one for which the assumptions are fulfilled. Note also that the initial point condition (A4) is needed in the proof, but in all practical experiments (see below), we can generalize compositionally without explicit knowledge of that point. We relegate further details to Appendix A.

## 4 Experiments

We validate our theoretical framework on the multi-sprite data. All models were trained for 2000 epochs on training sets of 100k samples using an NVIDIA RTX 2080 Ti; all test sets contain 10k samples. Table 1 summarizes the reconstruction quality achieved on the in-domain (ID) test set ($P$) and the entire latent space ($Q$) for all experiments.

**Motivating experiment** We implement the setup from Figure 1 to demonstrate that a compositional model does indeed generalize if the conditions from Theorem 1 are met. We model the component functions as four fully-connected layers followed by four upsampling-convolution stages, mapping the 5d component latent to $64 \times 64$ RGB images. For training stability, the composition function is implemented as a soft pixel-wise addition using the sigmoid function $\sigma(\cdot)$ as

$$\boldsymbol{x} = \sigma(\tilde{\boldsymbol{x}}_1) \cdot \tilde{\boldsymbol{x}}_1 + \sigma(-\tilde{\boldsymbol{x}}_1) \cdot \tilde{\boldsymbol{x}}_2, \tag{8}$$

which allows component 1 to occlude component 2. We contrast this to a non-compositional *monolithic* model, which has the same architecture as a single component function (with adjusted layer sizes to match the overall parameter count of the compositional model). Both models are trained on samples $(\boldsymbol{z}, \boldsymbol{x})$ from the training set using an MSE reconstruction loss. We show that both models have the capacity to fit the data by training on random samples covering the entire latent space (Table 1, **#1,2**). We then train on a distribution with orthogonal support as in equation 2, albeit with two planes for the foreground component to satisfy the sufficient support condition (Definition 3) as explained in Figure 4. Both models can reconstruct ID samples, but only the compositional model generalizes to the entire latent space (Table 1, **#3,4**).

**Flexible compositional support** Next, we demonstrate the variety of settings that fulfill the compositional support assumption as illustrated in Figure 3B and C. To this end, we repeat the experiment on training sets $P$ sampled from (i) a normal distribution with orthogonal support (Table 1, **#5**) and (ii) a uniform distribution over a diagonal support chosen broad enough to satisfy the sufficient support condition (Table 1, **#6**; see also Appendix C for details on how the support was chosen). The model generalizes to the entire latent space in both settings. Since the generalization performance is already close to the performance ceiling, broadening the support of both distributions (Table 1, **#7,8**) does not further increase performance.

**Violating Conditions** Finally, we look at the effect of violating some conditions.

- **Gaps in support** (Table 1, **#9**) Gaps in the support of the training set such that some component configurations are never observed (Figure 3E) violate the compositional support condition (Definition 2). While the overall reconstruction performance only drops slightly, visualizing the reconstruction error over a 2d-slice of the latent space in Figure 5 illustrates clearly that generalization fails exactly where the condition is violated.

- **Insufficient training variability** (Table 1, **#10**) Reducing the width of the diagonal support violates the sufficient support condition (Definition 3) as soon as some parts of the background component are always occluded and can not be observed in the output anymore (Compare Appendix C for details). We can clearly see that reconstruction performance on the entire latent space drops significantly as a result.

- **Collapsed Composition Function** (Table 1, **#11**) Changing the output of each component function from RGB to RGBa and implementing the composition as alpha compositing yields a model that is still compositional, but for which no support can satisfy the sufficient support condition since

| # | Train Set | Model | $R^2$ **ID** $\uparrow$ | $R^2$ **all** $\uparrow$ | $\Delta R^2$ $\downarrow$ |
|---|-----------|-------|-----------|-----------|-----------|
| 1 | Random | Monolithic | $0.931_{\pm 5.8e-4}$ | $0.931_{\pm 5.8e-4}$ | $0.000_{\pm 8.2e-4}$ |
| 2 | Random | Compositional | $0.957_{\pm 1.0e-3}$ | $0.957_{\pm 1.0e-3}$ | $0.000_{\pm 1.4e-3}$ |
| 3 | Orthogonal | Monolithic | $0.948_{\pm 1.7e-3}$ | $-0.500_{\pm 6.7e-2}$ | $1.448_{\pm 6.7e-2}$ |
| 4 | Orthogonal | Compositional | $0.957_{\pm 6.4e-4}$ | $0.951_{\pm 1.4e-3}$ | $0.006_{\pm 1.5e-3}$ |
| 5 | Ortho. $\sim \mathcal{N}$ | Compositional | $0.957_{\pm 5.5e-4}$ | $0.951_{\pm 1.0e-3}$ | $0.006_{\pm 1.1e-3}$ |
| 6 | Diagonal | Compositional | $0.954_{\pm 5.4e-3}$ | $0.945_{\pm 1.6e-2}$ | $0.009_{\pm 1.7e-2}$ |
| 7 | Ortho. (broad) | Compositional | $0.959_{\pm 9.7e-4}$ | $0.954_{\pm 1.3e-3}$ | $0.005_{\pm 1.6e-3}$ |
| 8 | Diag. (broad) | Compositional | $0.957_{\pm 1.2e-3}$ | $0.955_{\pm 1.2e-3}$ | $0.002_{\pm 1.7e-3}$ |
| 9 | Ortho. (gap) | Compositional | $0.954_{\pm 1.7e-3}$ | $0.895_{\pm 7.4e-3}$ | $0.059_{\pm 7.6e-3}$ |
| 10 | Diag. (narrow) | Compositional | $0.867_{\pm 7.5e-2}$ | $0.589_{\pm 2.8e-1}$ | $0.278_{\pm 2.9e-1}$ |
| 11 | Orthogonal | Comp. (RGBa) | $0.984_{\pm 1.8e-4}$ | $0.979_{\pm 1.4e-4}$ | $0.005_{\pm 2.3e-4}$ |

Table 1: We report the reconstruction quality measured as variance-weighted $R^2$ score (closer to 1 is better) on the in-domain (ID) test set and the entire latent space. As the ID region occupies a tiny fraction of the entire latent space, the difference in performance ($\Delta R^2$) indicates how well a model generalizes OOD. All results are averaged over 5 random seeds. **#1-4** The results demonstrate that a *monolithic* model fails to generalize in the setup from Figure 1, but a *compositional* model performs well on the entire latent space. **#5-8** Generalization can occur in a variety of settings that fulfill the sufficient conditions from Theorem 1. **#9,10** Violating the compositional and sufficient support condition prohibits generalization while choosing a more complex function class still works (**#11**). Table 2 in the appendix additionally reports the MSE for all experiments.

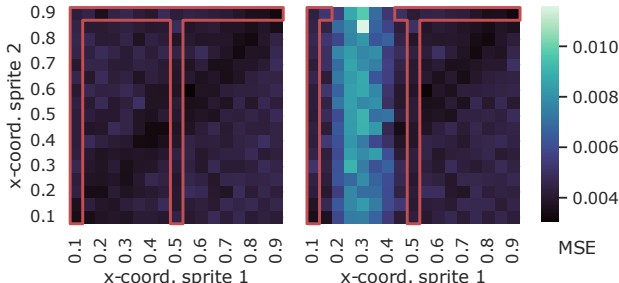

Figure 5: Heatmap of the reconstruction error over a $z_{1,x}$-$z_{2,x}$-projection of the latent space with overlaid training support (red). Generalization can occur when the support is compositional (left) but fails exactly where the support is incomplete at $z_{1,x} \in [0.14, 0.46]$ (right).

the derivative of transparent pixels will always be zero and the Jacobian matrix can therefore never have full rank (see Appendix B.1 for more details). However, we observe that the model still generalizes to the entire latent space and achieves even lower reconstruction error than the original model. This emphasizes that what we present are merely *sufficient* conditions. We include this experiment to motivate future work to find weaker conditions.

## 5 Discussion

We presented a first step and a framework to study compositional generalization in a more principled way. Clearly, there remain many open questions and limitations that we leave for future work.

**Supervised setting** We only studied a supervised regression setting in which the model has access to the ground-truth latents of each training sample. Or own findings underlined the results of previous works, e.g., Schott et al. [3] that compositional generalization is not trivial even in this setting. Ultimately, we are of course interested in the unsupervised setting akin to what is typically studied in identifiable representation learning. The unsupervised setting comes with inherent ambiguities as the

relationship between ground-truth latent space and inferred representations is unknown. No prior works exist that address this *identifiability* problem when training on a subset $P$ of the latent space, which makes generalizations guarantees as presented in this work harder to derive. Still, the results in this paper build an important foundation for future studies as sufficient conditions in the supervised setting can be considered necessary conditions in the unsupervised setting.

**Jacobian and initial point**   The proof of Theorem 1 utilizes the Jacobian of the ground-truth model. We emphasize again that this construction is necessary only for the proof and does not mean that we require access to the data-generating processes' full Jacobian for training. Similarly, the existence of an initial point $p^0$ is a technicality of the proof that is not reflected in the experiments. While it is not yet clear whether it is possible to complete the proof without the initial point condition, we believe there is a self-consistency condition that might alleviate the need for this condition. The experiments thus hint at the existence of alternative proof strategies with relaxed assumptions.

**Known composition function**   We also assume the composition function to be known which is approximately true in many interesting scenarios, such as object composition in scenes or the composition of instruments in music. In fact, many structured representation learning approaches like e.g. SlotAttention [18] incorporate structural components that are meant to mimic the compositional nature of the ground-truth-generating process. In other interesting cases like language, however, the composition function is unknown a priori and needs to be learned. This might be possible by observing how the gradients of $C$ change with respect to a fixed slot, at least if certain regularity conditions are fulfilled.

**Inductive biases**   Some of the conditions we derived can be relaxed in the presence of certain inductive biases. For example, models with an inductive bias towards shift invariance might be able to cope with certain gaps in the training support (e.g., if sprites are not visible in every position). Similarly, assuming all component functions $\varphi$ to be identical would substantially simplify the problem and allow for much smaller sufficient supports $P$. The conditions we derived do not assume any inductive bias but are meant to formally guarantee compositional generalization. We expect that our conditions generalize to more realistic conditions as long as the core aspects are fulfilled.

**Error bounds**   Our generalization results hold only if the learned model perfectly matches the ground-truth model on the training distribution. This is similar to identifiable representation learning, where a model must find the global minimum of a certain loss or reconstruction error for the theory to hold. Nonetheless, extending our results towards generalization errors that are bounded by the error on the training distribution is an important avenue for future work.

**Broader impact**   Compositional generalization, once achieved, has the potential to be benefit many downstream applications. By increasing sample and training efficiency, it could help to democratize the development and research of large-scale models. Better generalization capabilities could also increase the reliability and robustness of models but may amplify existing biases and inequalities in the data by generalizing them and hinder our ability to interpret and certify a model's decisions.

# 6   Conclusion

Machine learning, despite all recent breakthroughs, still struggles with generalization. Taking advantage of the basic building blocks that compose our visual world and our languages remains unique to human cognition. We believe that progress towards more generalizable machine learning is hampered by a lack of a formal understanding of how generalization can occur. This paper focuses on compositional generalization and provides a precise mathematical framework to study it. We derive a set of sufficient conditions under which compositional generalization can occur and which cover a wide range of existing approaches. We see this work as a stepping stone towards identifiable representation learning techniques that can provably infer and leverage the compositional structure of the data. It is certainly still a long road toward scalable empirical learning techniques that can fully leverage the compositional nature of our world. However, once achieved, there is an opportunity for drastically more sample-efficient, robust, and human-aligned machine learning models.

## Acknowledgments

We would like to thank (in alphabetical order): Jack Brady, Simon Buchholz, Attila Juhos, and Roland Zimmermann for helpful discussions and feedback.

This work was supported by the German Federal Ministry of Education and Research (BMBF): Tübingen AI Center, FKZ: 01IS18039A. WB acknowledges financial support via an Emmy Noether Grant funded by the German Research Foundation (DFG) under grant no. BR 6382/1-1 and via the Open Philantropy Foundation funded by the Good Ventures Foundation. WB is a member of the Machine Learning Cluster of Excellence, EXC number 2064/1 – Project number 390727645. This research utilized compute resources at the Tübingen Machine Learning Cloud, DFG FKZ INST 37/1057-1 FUGG. We thank the International Max Planck Research School for Intelligent Systems (IMPRS-IS) for supporting TW and PM.

## Author contributions

The project was led and coordinated by TW. TW and PM jointly developed the theory with insights from WB. TW implemented and conducted the experiments with input from PM and WB. TW led the writing of the manuscript with help from WB, PM, and MB. TW created all figures with comments from PM and WB.

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

# A Proof of Theorem 1

We reiterate the setup and notation introduced in the paper here for ease of reference.

**Notation** $[N]$ denotes the set of natural numbers $\{1, 2, ..., N\}$. Vector-valued variables (e.g., $\boldsymbol{x}$) and functions (e.g., $\boldsymbol{f}$) are written in bold. **Id** denotes the (vector-valued) identity function. We write the support of a distribution $P$ as $\operatorname{supp} P$. To express that two functions $\boldsymbol{f}, \boldsymbol{g}$ are equal for all points in the support of distribution $P$, i.e., $\boldsymbol{f}(\boldsymbol{x}) = \boldsymbol{g}(\boldsymbol{x}) \; \forall \boldsymbol{x} \in \operatorname{supp} P$, we write $\boldsymbol{f} \equiv_P \boldsymbol{g}$. Finally, $\frac{\partial \boldsymbol{f}}{\partial \boldsymbol{x}}$ denotes the total derivative of a vector-valued function $\boldsymbol{f}$ by all its inputs $\boldsymbol{x}$, corresponding to the Jacobian matrix with entries $\frac{\partial f_i}{\partial x_j}$.

**Setup** We are given two arbitrary distributions $P, Q$ over latents $\boldsymbol{z} = (\boldsymbol{z}_1, ..., \boldsymbol{z}_K) \in \mathcal{Z}$. Each latent $\boldsymbol{z}_k$ describes one of the $K$ *components* of the final data point $\boldsymbol{x}$ produced by the ground-truth data-generating process $\boldsymbol{f}$. A model $\hat{\boldsymbol{f}}$ is trained to fit the data-generating process on samples of $P$; the aim is to derive conditions on $P$ and $\hat{\boldsymbol{f}}$ that are sufficient for $\hat{\boldsymbol{f}}$ to then also fit $\boldsymbol{f}$ on $Q$.

We assume that $\boldsymbol{f}, \hat{\boldsymbol{f}}$ are chosen such that we can find at least one *compositional representation* (Definition 1) for either function that shares a common *composition function* $\boldsymbol{C}$ and factorization of the latent space $\mathcal{Z}_1 \times \cdots \times \mathcal{Z}_K = \mathcal{Z}$.

*Proof of Theorem 1.* For $\hat{\boldsymbol{f}}$ to generalize to $Q$, we need to show fitting $\boldsymbol{f}$ on $P$ implies also fitting it on $Q$, in other words

$$\boldsymbol{f} \underset{P}{\equiv} \hat{\boldsymbol{f}} \implies \boldsymbol{f} \underset{Q}{\equiv} \hat{\boldsymbol{f}} \tag{9}$$

*Step 1.* Since $\boldsymbol{C}$ is known and fixed, we immediately get

$$\boldsymbol{\varphi} \underset{Q}{\equiv} \hat{\boldsymbol{\varphi}} \implies \boldsymbol{f} \underset{Q}{\equiv} \hat{\boldsymbol{f}}, \tag{10}$$

i.e., it suffices to show that the *component functions* generalize. Note, however, that since $\boldsymbol{C}$ is not generally assumed to be invertible, we do *not* directly get that agreement of $\boldsymbol{f}, \hat{\boldsymbol{f}}$ on $P$ also implies agreement of their component functions $\boldsymbol{\varphi}, \hat{\boldsymbol{\varphi}}$ on $P$.

*Step 2.* We require $P$ to have *compositional support* w.r.t. $Q$ (Definition 2 and Assumption (A2)). The consequence of this assumption is that any point $\boldsymbol{q} = (\boldsymbol{q}_1, ..., \boldsymbol{q}_K) \in Q$ can be constructed from components of the $K$ *support points* $\boldsymbol{p}^k = (\boldsymbol{p}_1^k, ..., \boldsymbol{p}_K^k) \in P$ subject to $\boldsymbol{p}_k^k = \boldsymbol{q}_k$ as

$$\boldsymbol{q} = (\boldsymbol{p}_1^1, ..., \boldsymbol{p}_K^K). \tag{11}$$

A trivial consequence, then, is that points $\tilde{\boldsymbol{x}} \in \tilde{\mathcal{X}}$ in *component space* corresponding to points in $Q$ in latent space can always be mapped back to latents in $P$

$$\boldsymbol{\varphi}(\boldsymbol{q}) = (\boldsymbol{\varphi}_1(\boldsymbol{q}_1), ..., \boldsymbol{\varphi}_K(\boldsymbol{q}_K)) = \left(\boldsymbol{\varphi}_1\left(\boldsymbol{p}_1^{(1)}\right), ..., \boldsymbol{\varphi}_K\left(\boldsymbol{p}_K^{(K)}\right)\right) \tag{12}$$

because each *component function* $\boldsymbol{\varphi}_k$ only depends on the latents $\boldsymbol{z}_k$ of a single component. This is also the case for the component functions $\hat{\boldsymbol{\varphi}}$ of $\hat{\boldsymbol{f}}$ so that we get

$$\boldsymbol{\varphi} \underset{P}{\equiv} \hat{\boldsymbol{\varphi}} \implies \boldsymbol{\varphi} \underset{Q}{\equiv} \hat{\boldsymbol{\varphi}}. \tag{13}$$

*Step 3.* We now only need to show that $\boldsymbol{\varphi} \underset{P}{\equiv} \hat{\boldsymbol{\varphi}}$ follows from $\boldsymbol{f} \underset{P}{\equiv} \hat{\boldsymbol{f}}$. As noted above, this is not guaranteed to be the case, as $\boldsymbol{C}$ is not generally invertible (e.g., in the presence of occlusions). We, therefore, need to consider when a unique reconstruction of the component functions $\boldsymbol{\varphi}$ (and correspondingly $\hat{\boldsymbol{\varphi}}$) is possible, based on only the observations $\boldsymbol{x} = \boldsymbol{f}(\boldsymbol{z})$ on $Q$.

As explained in the main paper, we can reason about how a change in the latents $\boldsymbol{z}_k$ of some slot affects the final output, which we can express through the chain rule as

$$\underbrace{\frac{\partial \boldsymbol{f}}{\partial \boldsymbol{z}_k}(\boldsymbol{z})}_{N \times D} = \underbrace{\frac{\partial \boldsymbol{C}}{\partial \boldsymbol{\varphi}_k}(\boldsymbol{\varphi}(\boldsymbol{z}))}_{N \times M} \underbrace{\frac{\partial \boldsymbol{\varphi}_k}{\partial \boldsymbol{z}_k}(\boldsymbol{z}_k)}_{M \times D} \quad \forall k \in [K]. \tag{14}$$

Here, $N$ is the dimension of the final output (e.g., $64 \times 64 \times 3$ for RGB images), $M$ is the dimension of a component's representation $\tilde{x}_k$ (e.g., also $64 \times 64 \times 3$ for RGB images if composition happens in image space), and $D$ is the dimension of a component's latent description $z_k$ (e.g., 5: x-position, y-position, shape, size, hue for sprites). Note that we can look at the derivative component-wise because each *component function* $\varphi_k$ only depends on the latents $z_k$ of its component. However, the *combination function* still depends on the (hidden) representation of all components, and therefore $\frac{\partial C}{\partial \varphi_k}$ is a function of the entire $\varphi$ and $z$.

In Equation 14, the left-hand side (LHS) $\frac{\partial f}{\partial z_k}$ can be computed from the training, as long as $\operatorname{supp} P$ is an open set. On the right-hand side (RHS), the functional form of $\frac{\partial C}{\partial \varphi_k}$ is known since $C$ is given, but since $\varphi(z)$ is still unknown, the exact entries of this Jacobian matrix are unknown. As such, Equation 14 defines a system of partial differential equations (PDEs) for the set of component functions $\varphi$ with independent variables $z$.

Before we can attempt to solve this system of PDEs, we simplify it by isolating $\frac{\partial \varphi_k}{\partial z_k}$. Since all terms are matrices, this is equivalent to solving a system of linear equations. For $N = M$, $\frac{\partial C}{\partial \varphi_k}$ is square, and we can solve by taking its inverse as long as the determinant is not zero. In the general case of $N \geq M$, however, we have to resort to the pseudoinverse to write

$$\frac{\partial \varphi_k}{\partial z_k} = \left( \frac{\partial C}{\partial \varphi_k}^\top \frac{\partial C}{\partial \varphi_k} \right)^{-1} \frac{\partial C}{\partial \varphi_k}^\top \frac{\partial f}{\partial z_k} \quad \forall k \in [K], \tag{15}$$

which gives all solutions $\frac{\partial \varphi_k}{\partial z_k}$ if any exist. This system is overdetermined, and a (unique) solution exists if $\frac{\partial C}{\partial \varphi_k}$ has full (column) rank. In other words, to execute this simplification step on $P$, we require that for all $z \in P$ the $M$ column vectors of the form

$$\left( \frac{\partial C_1}{\partial \varphi_{km}}\big(\varphi(z)\big), ..., \frac{\partial C_N}{\partial \varphi_{km}}\big(\varphi(z)\big) \right)^\top \quad \forall m \in [M] \tag{16}$$

are linearly independent. Each entry of a column vector describes how all entries $C_n$ of the final output (e.g., the pixels of the output image) change with a single entry $\varphi_{km}$ of the intermediate representation of component $k$ (e.g., a single pixel of the component-wise image). It is easy to see that if even a part of the intermediate representation is not reflected in the final output (e.g., in the presence of occlusions, when a single pixel of one component is occluded), the entire corresponding column is zero, and the matrix does not have full rank.

To circumvent this issue, we realize that the LHS of Equation 15 only depends on the latents $z_k$ of a single component. Hence, for a given latent $z$ and a slot index $k$, the correct component function will have the same solution for all points in any (finite) set

$$P'(z, k) \subseteq \{p \in \operatorname{supp} P | p_k = z_k\}. \tag{17}$$

We can interpret these points as the intersection of $P$ with a plane in latent space at $z_k$ (e.g., all latent combinations in the training set in which one component is fixed in a specific configuration). We can then define a modified composition function $\tilde{C}$ that takes $z$ and a slot index $k$ as input and produces a "superposition" of images corresponding to the latents in the subset as

$$\tilde{C}(\varphi, z, k) = \sum_{p \in P'(z,k)} C\big(\varphi(p)\big). \tag{18}$$

Essentially, we are condensing the information from multiple points in the latent space into a single function. This enables us to write a modified version of Equation 14 as

$$\sum_{p \in P'(z,k)} \frac{\partial f}{\partial z_k}(p) = \sum_{p \in P'(z,k)} \frac{\partial C}{\partial \varphi_k}\big(\varphi(p)\big) \frac{\partial \varphi_k}{\partial z_k}(z_k) = \frac{\partial \tilde{C}}{\partial \varphi_k}(\varphi, z, k) \frac{\partial \varphi_k}{\partial z_k}(z_k) \quad \forall k \in [K] \tag{19}$$

Now we can solve for $\frac{\partial \varphi_k}{\partial z_k}$ as in Equation 15, but this time require only that $\frac{\partial \tilde{C}}{\partial \varphi_k}$ has full (column) rank for a unique solution to exist, i.e.,

$$\operatorname{rank} \frac{\partial \tilde{C}}{\partial \varphi_k}(\varphi, z, k) = \sum_{p \in P'(z,k)} \frac{\partial C}{\partial \varphi_k}\big(\varphi(p)\big) = M \quad \forall z \in P \quad \forall k \in [K]. \tag{20}$$

In general, this condition is easier to fulfill since full rank is not required in any one point but over a set of points. For occlusions, for example, any pixel of a slot can be occluded in some points $\boldsymbol{p} \in P'$, as long as it is not occluded in all of them. We can interpret this procedure as "collecting sufficient information" such that an inversion of the generally non-invertible $\boldsymbol{C}$ becomes feasible locally.

The requirement that $\operatorname{supp} P$ has to be an open set, together with the full rank condition on the Jacobian of the composition function condensed over multiple points, $\tilde{\boldsymbol{C}}$, is termed *sufficient support* in the main paper (Definition 3 and Assumption (A3)). As explained here, this means that the training distribution $P$, the composition function $\boldsymbol{C}$, and derivatives of the function $\boldsymbol{f}$ on the training set specify a unique relationship between the component function $\boldsymbol{\varphi}_k$ and its derivatives:

$$\frac{\partial \boldsymbol{\varphi}_k}{\partial \boldsymbol{z}_k} = \left( \frac{\partial \tilde{\boldsymbol{C}}}{\partial \boldsymbol{\varphi}_k}^\top (\boldsymbol{\varphi}, \boldsymbol{z}, k) \frac{\partial \boldsymbol{C}}{\partial \boldsymbol{\varphi}_k} \right)^{-1} \frac{\partial \boldsymbol{C}}{\partial \boldsymbol{\varphi}_k}^\top \sum_{\boldsymbol{p} \in P'(\boldsymbol{z}, k)} \frac{\partial \boldsymbol{f}}{\partial \boldsymbol{z}_k}(\boldsymbol{p}) \quad \forall k \in [K]. \tag{21}$$

As explained above, this solution to the linear system of equations constitutes a system of partial differential equations (PDEs) in the set of component functions $\boldsymbol{\varphi}$ with independent variables $\boldsymbol{z}$. We can see that this system has the form

$$\partial_i \boldsymbol{\varphi}(\boldsymbol{z}) = \boldsymbol{a}_i(\boldsymbol{z}, \boldsymbol{\varphi}(\boldsymbol{z})), \tag{22}$$

where $i \in [L]$ with $L := KD$ is an index over the flattened dimensions $K$ and $D$ such that the system of PDEs contains the functional relation between $\boldsymbol{\varphi}$ and its derivative from Equation 21 for all values of $k$. $\boldsymbol{a}_i$ is the combination of corresponding terms from the RHS of Equation 21. If this system of PDEs allows for more than one solution, Equation 21 does not uniquely determine the component functions.

However, if we have access to some initial point for which we know $\boldsymbol{\varphi}(\boldsymbol{0}) = \boldsymbol{\varphi}^0$, we can write

$$\begin{aligned}
\boldsymbol{\varphi}(z_1, ..., z_L) - \boldsymbol{\varphi}^0 = & \big( \varphi(z_1, ..., z_L) - \varphi(0, z_2, ..., z_L) \big) \\
& + \big( \varphi(0, z_2, ..., z_L) - \varphi(0, 0, z_3, ..., z_L) \big) \\
& + ... \\
& + \big( \varphi(0, ..., 0, z_L) - \varphi(0, ..., 0) \big).
\end{aligned} \tag{23}$$

In each line of this equation, only a single $z_i =: t$ is changing; all other $z_1, ..., z_L$ are fixed. Any solution of Equation 23, therefore, also has to solve the $L$ ordinary differential equations (ODEs) of the form

$$\partial_t \boldsymbol{\varphi}(z_1, ..., z_{i-1}, t, z_{i+1}, ..., z_L) = \boldsymbol{a}_i(z_1, ..., z_{i-1}, t, z_{i+1}, ..., z_L, \boldsymbol{\varphi}(z_1, ..., z_{i-1}, t, z_{i+1}, ..., z_L)), \tag{24}$$

which have a unique solution if $\boldsymbol{a}_i$ is Lipschitz in $\boldsymbol{\varphi}$ and continuous in $z_i$, as guaranteed by (A1). Therefore, 23 has at most one solution. This reference point does not have to be in $\boldsymbol{z} = \boldsymbol{0}$, as a simple coordinate transform will yield the same result for any point in $P$. It is therefore sufficient that there exists *some* point $\boldsymbol{p}^0 \in P$ for which $\boldsymbol{\varphi}(\boldsymbol{p}^0) = \hat{\boldsymbol{\varphi}}(\boldsymbol{p}^0)$ to obtain the same unique solution for $\boldsymbol{\varphi}$ and $\hat{\boldsymbol{\varphi}}$, which is exactly what (A4) states. Overall, this means that Equation 21 does indeed specify unique component functions $\boldsymbol{\varphi}_k$.

To recap, we have shown that the training distribution $P$, the composition function $\boldsymbol{C}$, and knowledge of $\boldsymbol{f}(\boldsymbol{z})$ for $\boldsymbol{z} \in \operatorname{supp} P$ specifies a unique relationship between the component functions $\boldsymbol{\varphi}_k$ and their derivative (Equation 21). With additional knowledge of one initial point, this constrains the problem enough to get a unique solution for the component functions $\boldsymbol{\varphi}_k$. Since $P$ and $\boldsymbol{C}$ are fixed, this means that if the functions $\boldsymbol{f}$ and $\hat{\boldsymbol{f}}$ agree on $P$, they specify the same component functions, i.e.,

$$\boldsymbol{f} \underset{P}{\equiv} \hat{\boldsymbol{f}} \implies \boldsymbol{\varphi} \underset{P}{\equiv} \hat{\boldsymbol{\varphi}} \tag{25}$$

*Step 4.* Finally, we can conclude the model $\hat{\boldsymbol{f}}$ fitting the ground-truth generating process $\boldsymbol{f}$ on the training distribution $P$, through Equations 25, 13, and 10, implies that the model generalizes to $Q$. In other words, Equation 9 holds.

$\square$

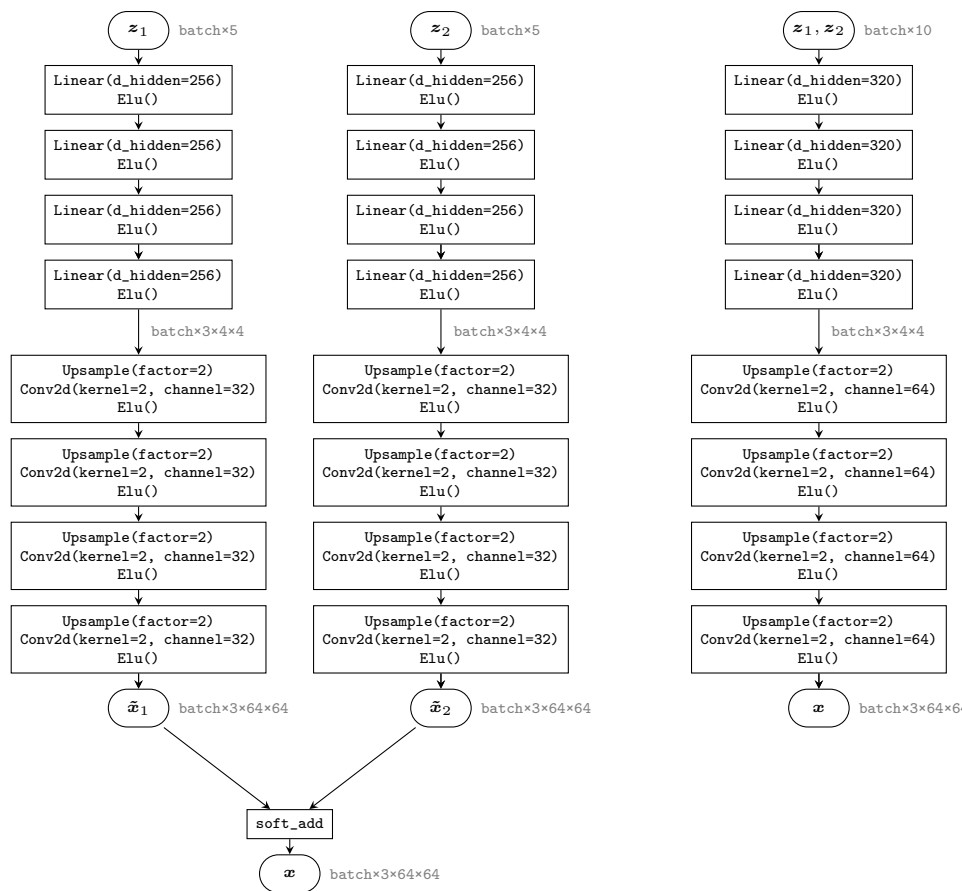

Figure 6: **Schematic of the employed models**. The monolithic model (right) uses the same architecture as each component model of the compositional model (left), except with a higher number of hidden units and channels to match the number of parameters.

## B  Experimental Details

Figure 6 shows a schematic of the employed compositional and monolithic model from Section 4.

Table 2 additionally reports the reconstruction quality measures as the mean squared error (MSE) for the experiments from Section 4.

### B.1  Details about the compositional functions

As explained in Equation 8 in section 4, the composition function is implemented as a soft pixel-wise addition in most experiments. The use of the sigmoid function $\sigma(\cdot)$ in the composition

$$\boldsymbol{x} = \sigma(\tilde{\boldsymbol{x}}_1) \cdot \tilde{\boldsymbol{x}}_1 + \sigma(-\tilde{\boldsymbol{x}}_1) \cdot \tilde{\boldsymbol{x}}_2 \tag{26}$$

was necessary for training stability. With this formulation, sprites can also overlap somewhat transparently, which is not desired and leads to small reconstruction artifacts for some specific samples. Implementing the composition with a step function as

$$\boldsymbol{x} = \text{step}(\tilde{\boldsymbol{x}}_1) \cdot \tilde{\boldsymbol{x}}_1 + \text{step}(-\tilde{\boldsymbol{x}}_1) \cdot \tilde{\boldsymbol{x}}_2 \tag{27}$$

instead would be more faithful to the ground-truth data-generating process, but is hard to train with gradient descent.

Note that both formulations could easily be extended to more than one sprite by simply repeating the composition operation with any additional sprite.

| # | Train Set | Model | MSE ID ↓ | MSE all ↓ |
|---|-----------|-------|----------|-----------|
| 1 | Random | Monolithic | $1.73e{-}3_{\pm1.5e-5}$ | $1.73e{-}3_{\pm1.5e-5}$ |
| 2 | Random | Compositional | $1.07e{-}3_{\pm2.6e-5}$ | $1.07e{-}3_{\pm2.6e-5}$ |
| 3 | Orthogonal | Monolithic | $8.49e{-}4_{\pm2.9e-5}$ | $4.06e{-}2_{\pm3.9e-3}$ |
| 4 | Orthogonal | Compositional | $6.94e{-}4_{\pm1.1e-5}$ | $1.24e{-}3_{\pm4.1e-5}$ |
| 5 | Ortho.$\sim \mathcal{N}$ | Compositional | $7.01e{-}4_{\pm8.7e-6}$ | $1.24e{-}3_{\pm2.6e-5}$ |
| 6 | Diagonal | Compositional | $8.87e{-}4_{\pm1.0e-4}$ | $1.39e{-}3_{\pm4.0e-4}$ |
| 7 | Ortho. (broad) | Compositional | $6.50e{-}4_{\pm1.5e-5}$ | $1.16e{-}3_{\pm3.3e-5}$ |
| 8 | Diag. (broad) | Compositional | $9.51e{-}4_{\pm2.8e-5}$ | $1.13e{-}3_{\pm3.1e-5}$ |
| 9 | Ortho. (gap) | Compositional | $7.36e{-}4_{\pm2.7e-5}$ | $2.64e{-}3_{\pm1.8e-4}$ |
| 10 | Diag. (narrow) | Compositional | $2.22e{-}3_{\pm1.2e-3}$ | $1.04e{-}2_{\pm7.1e-3}$ |
| 11 | Orthogonal | Comp. (RGBa) | $2.67e{-}4_{\pm2.8e-6}$ | $5.24e{-}4_{\pm3.7e-6}$ |

Table 2: We report the reconstruction quality measured as mean squared error (MSE, lower is better) for both the in-domain (ID) test set and the entire latent space, averaged over 5 random seeds.

In section 4, we also looked at a model that implements the composition through alpha compositing instead (see also Table 1, **#11**). Here, each component's intermediate representation is an RGBa image. The components are then overlaid on an opaque black background using the composition function

$$x_\alpha = x_{1,\alpha} + (1 - x_{1,\alpha}) \cdot x_{2,\alpha} \tag{28}$$

$$x_{\text{RGB}} = x_{1,\alpha} \cdot x_{1,\text{RGB}} + (1 - x_{1,\alpha}) \cdot \frac{x_{2,\alpha}}{x_\alpha} \cdot x_{2,\text{RGB}}. \tag{29}$$

While this yields a compositional function, the sufficient support condition (Definition 3) is generally not fulfilled on the sprites data. The reason is that in fully transparent pixels ($\alpha = 0$), changing the RGB value is not reflected in the output. Conversely, if a pixel is black, changing its alpha value will not affect how it is blended over a black background. As a result, most columns in the Jacobian $\frac{\partial C}{\partial \varphi_k}$ (see also Equation 16) will be zero. Since the intermediate representations of each sprite will contain a lot of black or transparent pixels (the entire background), the rank of the Jacobian here will be low. In this case, the workaround from Equation 18 does not help since the low rank is not a result of another component in the foreground but of the specific parameterization of each component itself.

As stated in the main paper, the fact that this parameterization still produces good results and generalizes well is an indicator that there might be another proof strategy or workaround that avoids this specific issue.

## C Details on the sufficient support assumption

The *sufficient support assumption* as stated in Definition 3 amounts to a rank condition on the Jacobian of the final output $x$ by the intermediate component representation $\tilde{x}$: For each configuration of a given component, $P$ must be sufficiently large so that tracking the dependence of each output dimension on each dimension of the component representation is possible. Step 3 of the proof of Theorem 1 outlines how this condition arises in general: For some interactions between components, it is possible that the final output $x$ becomes invariant to changes in the representation $\tilde{x}_k$ of component $k$. We can assume that the output is not generally invariant to component $k$ (since the component could then just be dropped from the data-generating process) and instead only arises for certain configurations of components. We provide additional examples here that illustrate how such an invariance can arise for different choices of $C$ and detail how a sufficient support can be chosen in these cases. Note that while it is possible to calculate the sufficient support condition directly, this is costly to do on a dataset scale since it involves calculating large Jacobians for many different data points. Instead, it is helpful to analyze which specific component configurations would violate the condition and choose the training set accordingly.

**2d sprites with occlusion (orthogonal sampling)** As explained in the main text and illustrated in Figure 4, occlusions make the output image invariant to changes of the background sprite in the

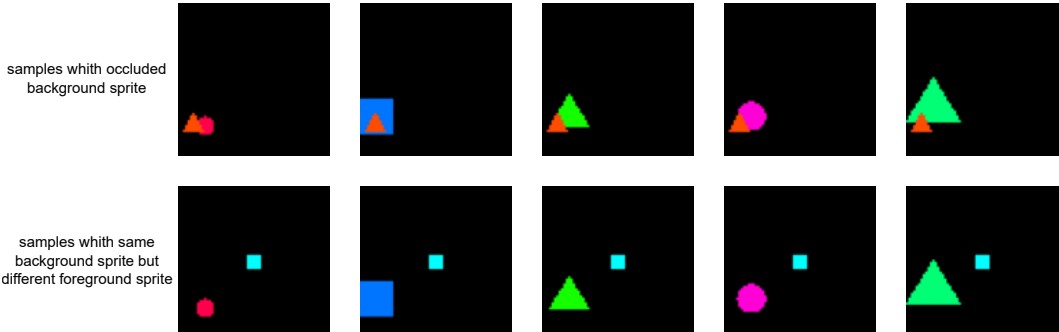

Figure 7: For each configuration of the background sprite, the training distribution must contain at least two samples in which the foreground sprites do not overlap the same pixels.

occluded pixels. The model does not receive a training signal for these pixels, and reconstruction becomes impossible. In the case of orthogonal sampling, the foreground sprite is fixed in its base configuration (orange triangle in the bottom left corner) whenever the background sprite is sampled randomly; see the first row of Figure 7. The foreground sprite, therefore, always occludes the same background pixels, for which the Jacobian $\frac{\partial C}{\partial \varphi_{km}}$ is zero. Therefore, for each configuration of the background sprite, a second data point is required in which the foreground sprite occludes a different and distinct set of pixels (cyan square in the second row of Figure 7). We can clearly observe that this is the case here; therefore, the Jacobians $\frac{\partial C}{\partial \varphi_{km}}$ from both samples have zero entries at different indices $m$ and their sum has full rank. The resulting sufficient support has the shape illustrated on the left of Figure 5.

**2d sprites with occlusion (diagonal sampling)**    As explained above, sufficient support for occluding sprites is guaranteed if the dataset contains at least two samples with the same background sprite and different foreground sprites that overlap distinct pixels (as also illustrated in Figure 4). In the diagonal sampling case, we, therefore, need to choose the width of the diagonal broad enough to guarantee that it contains such foreground sprites. The smallest possible width of the diagonal can be determined by finding the smallest possible x-offset (or y-offset) for which a pair of sprites of the smallest scale does not overlap. This is the case for a width of $0.2$, which was used for the *diagonal* sampling case in Table 1, **#6**. The *broad diagonal* sampling case Table 1, **#8** used a width of $0.4$ (double the minimal width), while the *narrow diagonal* sampling case in Table 1, **#10** used a width of $0.1$ (half the minimal width).

**Attributes and transformations of an object**    The sufficient support assumption is also necessary If the composition function $C$ is chosen to model the interaction of multiple attributes or transformations on a single object. For example, if one component models the rotation of an object and another models its shape, and the composition happens in pixel space, then a circular shape will be invariant to rotations. The Jacobian of the output by the rotation component will not have full rank whenever the shape component is circular. A sufficient support must contain at least one non-circular sample for each rotation angle in the training set. See Appendix D for an empirical observation of this phenomenon.

**Overlaying audio signals**    Sufficient support has to be guaranteed whenever the composition function contains $\max(\cdot)$ or $\min(\cdot)$ operations or similar operations with zero gradients. For example, if two or more audio signals are picked up by a microphone modeled with finite gain, a signal above a certain amplitude will saturate the microphone. The sufficient support assumption guarantees that for the configuration of one audio signal, there exists at least one sample in the dataset in which the other signals are below that threshold, such that changes in the signal are registered by the microphone.

**Vector or matrix products**    If the intermediate component representations $\tilde{x}$ are vectors or matrices and the composition function is their (outer) product, changes in the first vector (matrix) of the product do not change the output if the second vector (matrix) multiplies with a zero-element (or



Figure 8: **Composition of object attributes**. Each set of three figures shows a traversal of one component: `rotation`, `color`, and `blur`. The intermediate output of the first component function $\varphi_1$ is an image of a white square at a specific angle; the intermediate output of the other two component functions are convolution filters. Note that for strong Gaussian blurs, the shape becomes circular, making the output invariant to rotations.

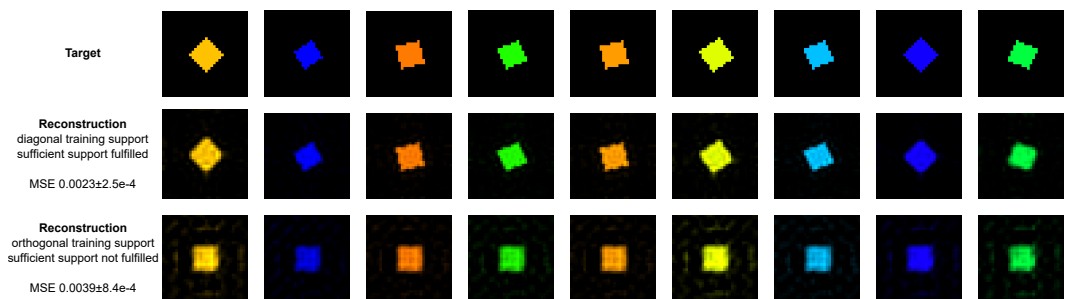

Figure 9: **Generalizing to novel compositions of object attributes**. For the data-generating process illustrated in Figure 8, a compositional model can learn to generalize to unseen compositions of `rotation`, `color`, and `blur` when all conditions are fulfilled (middle row). If sufficient support is violated, in this case due to the rotation invariance introduced by the Gaussian blur, the model is unable to learn the rotation transformation. he reported OOD MSE is calculated for samples with `blur = 0`.

zero-row/column). The sufficient support assumption guarantees that the training set contains some configuration of the second vector (matrix) with zero elements at different locations.

# D   Additional experiments

## D.1   Composition of Object Attributes

To demonstrate that the considered function class does not only apply to the composition of *objects* but can also model the composition of *object attributes*, we consider a modification of our sprite setting, in which each image contains a single sprite with configured by the three attributes `rotation`, `color`, and `blur`. Figure 8 illustrates the resulting data. We chose these attributes because, as outlined in the previous section, they illustrate the connection between the sufficient support assumption and transformation invariances. Additionally, these attributes can be implemented with a simple composition operation: convolution. The resulting generating process can, thus, be formulated as

$$\boldsymbol{f}(\boldsymbol{z}) = \boldsymbol{C}(\boldsymbol{\varphi}(\boldsymbol{z})) = \boldsymbol{\varphi}_1(\boldsymbol{z}_1) * \boldsymbol{\varphi}_2(\boldsymbol{z}_2) * \boldsymbol{\varphi}_3(\boldsymbol{z}_3), \tag{30}$$

where $\varphi_1$ renders the sprite with a specific rotation, $\varphi_2$ generates a convolution filter that changes the color, and $\varphi_3$ generates a Gaussian blur filter.

As shown in Figure 8, the sprite's appearance is invariant to rotations whenever a strong Gaussian blur is applied. Consequently, to fulfill the sufficient support condition, the support of the training distribution needs to be chosen such that rotations are observed on configurations for which the blur is not too strong. To demonstrate this, we train a compositional model on two different training distributions: The first one has a diagonal support as in Experiment 1 #6, which fulfills the *sufficient support* assumption. The second one has an orthogonal support as in Experiment 1 #3,4 chosen such that different rotations are only observed with a strong Gaussian blur. Figure 9 shows reconstructions of random OOD samples for both models. While the model learns to reconstruct the color correctly on both training sets, the rotation can only be learned in the first case when sufficient support is fulfilled.

| Train Set | $R^2$ **ID** $\uparrow$ | $R^2$ **OOD** $\uparrow$ | $\Delta R^2$ $\downarrow$ |
|---|---|---|---|
| Orthogonal | $0.676_{\pm 4.8e-3}$ | $0.632_{\pm 2.9e-3}$ | $0.044_{\pm 5.6e-3}$ |
| Orthogonal with gap | $0.560_{\pm 1.5e-2}$ | $0.345_{\pm 1.3e-2}$ | $0.215_{\pm 2.0e-2}$ |

Table 3: **Experiment results on CLEVR** [35]. On an orthogonal training set (Equation 31) that fulfills the assumptions from Theorem 1, a compositional model is able to generalize compositionally, as indicated by the minuscule difference in performance on ID and OOD points. When the compositional support assumption is violated by introducing a gap in the support of one latent, generalization fails, and the difference between ID and OOD performance increases significantly. The overall lower ID performance on the training set with a gap (and the corresponding proportionally lower OOD performance) are due to the smaller number of samples in this training set, as explained below. Results are averaged over three seeds.

## D.2 Compositional generalization on CLEVR

We additionally conduct experiments on the CLEVR dataset [35], a popular benchmark for compositional generalization and object-centric learning. The original CLEVR dataset consists of simple 3d scenes containing a varying number of objects configured by their x-position, y-position, shape, color, size, and orientation. While CLEVR comes with the code to generate new samples, generation cannot be controlled precisely: Objects are always positioned randomly, no specific combinations of object attributes other than the relation between shape and color can be controlled, and collisions are always avoided. As a result, we cannot easily generate large, controlled training sets with precisely defined supports and distributions as in Section 4 and instead opt to filter the existing dataset.

Specifically, we select all 13145 images containing exactly three objects (since Theorem 1 implicitly assumes that the number of slots is known). We filter an orthogonal training set similar to the one described in Equation 2, except that we have three slots and replace equality to the base configurations with a distance measure $d$, i.e.,

$$\text{supp} \, P = \left\{ (\boldsymbol{z}_1 \in \mathcal{Z}_1, \boldsymbol{z}_2 \in \mathcal{Z}_2, \boldsymbol{z}_3 \in \mathcal{Z}_3) | d(\boldsymbol{z}_1, \boldsymbol{z}_1^0) < \delta \vee d(\boldsymbol{z}_2, \boldsymbol{z}_2^0) < \delta \vee d(\boldsymbol{z}_3, \boldsymbol{z}_3^0) < \delta \right\}. \tag{31}$$

This modification is necessary since barely any two samples use the exact same configurations of an object. For our experiments, we use $\delta = 0.495$ and set aside $10\,\%$ of the ID samples for evaluation. We end up with 4523 ID training samples, 502 ID test samples, and 8120 OOD test samples.

We train a compositional model as outlined in Section 4 and shown in Figure 6, except that we add an additional upsampling and convolution layer to get to a final output size of $128 \times 128 \times 3$.

The results are summarized in Table 3: On the orthogonal training set that satisfies the assumptions from Theorem 1, ID and OOD performance are nearly identical, indicating that the model is able to generalize OOD. However, if we introduce a gap in the support of $\boldsymbol{z}_1$ as in Experiment **#9** in Section 4, we see that the gap between ID and OOD performance increases significantly, showing that violating the compositional support condition prohibits generalization. Note that since the ID and OOD sets are filtered from a fixed pool of samples, introducing this gap in the ID set effectively reduces the size of the ID training and test sets to 3254 and 361, respectively. The drop in ID performance and proportional decrease in OOD performance can be attributed to the reduced size of the training set.

