# OpenReview forum: "Compositional Generalization from First Principles"
_NeurIPS.cc/2023/Conference — NeurIPS 2023 poster_

### Official Review · Reviewer_1tYF · 2023-06-08

**Soundness:** 3 good
**Presentation:** 3 good
**Contribution:** 3 good
**Rating:** 5
**Confidence:** 3

**Summary:**

This paper theoretically investigates mathematical conditions for compositional generalization.
It defines compositional representation and compositional support, then derives a set of sufficient conditions on training distribution support and model architecture.
It also conducts empirical validation and ablation and demonstrates the application of the theory in real-world scenarios.

**Strengths:**

The lack of compositional generalization ability has been a longstanding problem, and it is good to have a theory for it.

- The definitions and the theorem are clearly stated.

- The theorem has a general form, indicating potential wide application.

- The theorem is derived strictly.

- The theorem covers a wide range of existing approaches.

- The running example helps to explain the concepts and ideas.

- The experiments validate the theoretical results.

**Weaknesses:**

The setting is strong so the theorem is weak and only covers limited cases.
Although this is an early work, it is still good to have wide coverage.

(1) The input is disentangled factors, but many problems have entangled inputs.

(2) It is not discussed whether the theorem applies to parametric composition function C.
Though the generative processes are known in many interesting scenarios (line 272), we may not know the exact parametric values in the process, e.g., generating a human face from face parts.
In such a case, we may treat the composition process as a parametrized model, and the parameters are learned during training.
However, the parameters may fit the training data and not generalize to test data.
It means equation (9) in the proof does not naturally hold.

(3) It would be helpful to have an experiment on a complicated problem that fits the settings and satisfies the conditions.

**Questions:**

Please respond to the weakness section.

(4) It might need a check for line 68: ``The OOD generalization ... where train and test distributions differ in their densities, but not their supports ...``.
If OOD means "Out-of-distribution," it seems to treat disjoint supports.

**Limitations:**

The limitations are mentioned mainly in the discussion section.

---

> ### Author Rebuttal · Authors · 2023-08-09
>
> We thank the reviewer for reading our work and for commending the clarity of work. We encourage the reviewer to check our global response where we have addressed some of their concerns, nonetheless we address some specific points here:
>
> 1. (Weakness 0: “The theorem is weak and only covers limited cases”) We agree that the theorem does not encompass all possible formulations of compositional generalization. However, **we hope to have clarified in points 1-4 of the global response that our theory still covers many different settings in various domains**. As also outlined there in the final paragraph of the global response, compared to earlier works on the same problem (chiefly [1, 2]) our assumptions on the composition function are much weaker and cover many more interesting cases. Diverging from these mentioned approaches, our work distinguishes itself by providing sufficient conditions that enable compositional generalization of a broader function class, while making minimal assumptions on the train and test support.
> 2. (Weakness 1) To enhance clarity for our readers, we employ the multi-sprite setting featuring 'disentangled' factors as a means to elucidate our methodology. A our theory makes no assumptions on the relationship of latent factors across or within each slot, our theorem is valid for inputs that contain ‘entangled’ factors. What’s more, **the diagonal sampling case in our experiments demonstrates that our theorem holds for highly correlated inputs**.
> 3. (Weakness 2) **Our theorem does apply to differentiable parametric functions C that remain fixed across the entire domain**. In practice, as the reviewer correctly notices, training a parametric composition function on only a subset of the entire domain will make it difficult to guarantee its behavior outside the training set. Even so, for specific parameterizations of C this might be possible; we are in fact exploring this direction in a follow up to this work. Besides that we want to stress again that prior work [1] also assumes knowledge of $C$ and additionally only allows $C$ to be from a very limited function class. **Our work therefore constitutes a considerable step forward from this methods.**
> 4. (Weakness 3): **We have conducted an additional experiment** with different component and composition functions to further demonstrate the generality of our theorem. For more details we kindly direct the reviewer's attention to points 8 and 9 in the global response and the accompanying rebuttal document. As highlighted in the concluding section of our global response, we kindly ask to judge the value of our contribution based on the theoretical results which make up the bulk of our work and greatly improve upon prior works. The application of our results to more complicated real-world datasets which are substantially harder to analyze is a direction we intend to explore in subsequent works.
> 5. (Question 1): The reviewer is correct that a better distinction between out-of-distribution (OOD) and out-of-support settings (OOS) is needed. In our work, we use the term OOD to encompass both OOS settings and settings where the densities differ, but not the support (for lack of a better term, we call this out-of-density here). In L68, we meant to say many prior works study out-of-density settings but not out-of-support settings. Our setting encompasses both, but compositional generalization clearly becomes trivial in out-of-density settings. **We have rephrased the section to better explain this distinction.**
>
> [1] https://arxiv.org/abs/2211.11719

---

> > ### Comment · Reviewer_1tYF · 2023-08-12
> >
> > Thank you for answering the questions.
> > I have some further questions.
> >
> > (Weakness 1)
> > The model seems to have ground-truth latent factors (z1, z2) as an input and an image x as an output for the multi-sprite setting (line 133 and line 221).
> > So it has disentangled inputs (z1, z2) and entangled outputs x.
> > So the review says it is not designed for entangled model inputs.
> >
> > Also, I am unsure whether Figure 1 may be confusing, as it seems to indicate that inputs are images.
> >
> > Also, in 259, "We only studied a supervised regression setting in which the model has access to the ground-truth latents of each training sample."
> > If the inputs are latents, does the method also require ground-truth latents of each test sample?
> >
> >
> > (Weakness 2)
> > A core problem of compositional generalization is to make a parametric model work well outside the training domain.
> > So it seems to be a strong requirement that if C is a parametric function, it needs to "remain fixed across the entire domain."
> > Readers may expect that C is a general parametric model, e.g., a neural network, so clarifying the property of C may be helpful.

---

> > > ### Author Response · Authors · 2023-08-14
> > >
> > > We thank the reviewer for reaching out for further clarification.
> > >
> > > We believe there might still be a misunderstanding about the training setting. Throughout the entirety of our work, we consider fitting a regression model which takes component descriptions $\boldsymbol z$ as input and outputs a composition $\hat{\boldsymbol x} = \boldsymbol f(\boldsymbol z)$, as illustrated in the left of the updated Figure 1 and the structural diagram in Figure 2.
> > >
> > > - In training, each sample is a tuple $(\boldsymbol z, \boldsymbol x)$, and the loss is calculated between the model’s reconstruction $\hat{\boldsymbol x}$ and the target $\boldsymbol x$. **We adapted the experiment section to now state this explicitly**. For Figure 1, this means that the *target* images outlined in blue are part of the training set, just as labels in, say, a classification setting would be. **We updated the caption of Figure 1** to include the phrase “regression model”, as we believe that the term “target” is understood well enough within a regression setting to avoid any further confusion.
> > > - During testing, the model is given an input $\boldsymbol z$ and outputs an output image $\hat{\boldsymbol x}$. The reviewer is correct in interpreting L.259 to mean that.
> > >
> > > We believe part of the confusion might be our choice to refer to the model’s input as latent factors. We chose this terminology as it is common when talking about the inputs of data-generating processes and **now include a clarification that these latents are not inferred from the data** as would be the case in an unsupervised setting. We maintain that the regression problem alone is an interesting and challenging problem.
> > >
> > > (Weakness 1) The reviewer is correct in saying that the model expects ordered, disentangled inputs. We originally understood the reviewer’s question about entangled latents to mean that $\boldsymbol z_1, \boldsymbol z_2$ share a common factor or are highly correlated, which would be covered by our theory as explained above.
> > >
> > > (Weakness 2) We remind the reviewer that the $\boldsymbol\varphi_k$ are also arbitrary parametric functions so that the model can be expressive even with rather simple $\boldsymbol C$, which is trivially fixed across $Q$. Nonetheless, **we now include a note about parametric $\boldsymbol C$ in the discussion section**.

---

> > > > ### Author Response · Authors · 2023-08-20
> > > >
> > > > We hope to have satisfactorily addressed all the reviewer’s questions. We also want to turn the reviewer’s attention to the comment section with reviewer yWdt, where we discuss different training settings and agree to look into additional experiments for the camera-ready; this should further resolve the reviewer’s original point 3.
> > > >
> > > > With the rebuttal period coming to an end and most of the reviewer’s original concerns addressed, we hope that the reviewer’s own assessment still holds that our work deals with **“a longstanding problem”** for which **“it is good to have a theory”**, especially one in a **“general form, indicating potential wide application”** and that can **“cover a wide range of existing approaches”**. Combined with the views of the other reviewers that **“the paper studies a very important problem from a unique perspective”** (5Ysh) and promises to be **“valuable for future work related to compositional generalization”** (yWdt), we hope that the reviewer can agree that our work will be of interest to the community and can therefore confidently adjust the score and recommend the paper’s acceptance.

---

> > > > > ### Comment · Reviewer_1tYF · 2023-08-21
> > > > >
> > > > > Thank you for the rebuttal. I keep the score.

---

### Official Review · Reviewer_5Ysh · 2023-06-23

**Soundness:** 3 good
**Presentation:** 3 good
**Contribution:** 3 good
**Rating:** 6
**Confidence:** 4

**Summary:**

Compositional generalization, which is easy for humans but hard for deep neural networks, gradually attracts more attention. Different from most existing works focusing on how to train a more to compositionally generalize well, this paper aims at the characteristics of the datasets. It tries to find sufficient conditions for the dataset to make compositional generalization possible. Specifically, the paper claims that the compositional data-generating process is the key to compositional generalization. It also proves a set of sufficient conditions under which models trained on the data are able to generalize compositionally. The proposed theory is verified by a manually designed vision dataset.

The paper studies a very important problem from a unique perspective, which is novel to me. The theoretical analysis also has the potential to be extended to more general scenarios. However, I do have some concerns and confusion about the current version. I wish the authors could clarify some of them during the rebuttal phase. So I would give the current version a borderline accept, and I would be happy to increase my score if my following concerns are resolved.

**Strengths:**

See the summary part.

**Weaknesses:**

Based on the definition in section 3.1 and the illustrative example in Figure 2, it seems that the compositional data generation needs multiple objects. For example, in equation 1, each $\phi_i(z)$ determines one object and function $C$ is used to combine them into a single input signal (like putting different objects on the same image). However, different $\phi_i(z)$ can also be different parts or different characteristics of ONE object. For example, we can use $\phi_1(z)$ and $\phi_2(z)$ to represent the silhouette and fillings of the same object (then $z$ should be some low-level semantic concepts, like different edges for silhouette and texture + color for the fillings). My main concern here is that in practice, the form of compositionality could be various, not only for different objects (parts) on the same input signal. The theoretical framework proposed in this paper also has the potential to cover those more general compositional relationships. The example proposed in the paper narrows down the scope of the theory. I think it would be very helpful to broaden the scope by providing extra examples that are quite different from the sprites one (e.g., an NLP example) but can still be explained by the theorem. This will strengthen the paper a lot.

**Questions:**

1. There is one citation missing (i.e., the ? mark) in the introduction part.

2. It is easy to misunderstand Figure 1. It would be better to align the left and right panels and directly put the subtitle of different rows on the figure, e.g., row 3 contains images generated by the model, and row 4 contains the targets.
3. In the second paragraph of section 3.1, the subscript should be $z_{2,x}$
4. What are $z_2^0$ and $z_1^0$ in equation 2?
5. Figures 3 and 4 use similar styles, but I guess they are discussing different things. In Figure 3, the x and y axis can be any attributes, e.g., position, color, etc., but those in Figure 4 seems only to refer to x and y positions.

**Limitations:**

1. For the experimental part, it would be helpful to demonstrate what the monolithic and compositional models look like (maybe in the appendix): it would be hard to imagine the difference between them without the diagrams.
2. IIUC, the models used in the experimental parts are vanilla deep neural networks. Showing results on some models (and algorithms) designed for disentanglement or compositional learning might strengthen the paper a lot. The emergent-communication-based methods [1,2] or VAE-based methods [3] are both good options.
3. The problem studied in [3] might be highly correlated with this paper.

[1] Xu, Zhenlin, Marc Niethammer, and Colin A. Raffel. "Compositional generalization in unsupervised compositional representation learning: A study on disentanglement and emergent language." *NeurIPS 2022*

[2] Ren, Yi, et al. "Compositional languages emerge in a neural iterated learning model." ICLR 2020

[3] Schott, Lukas, et al. "Visual representation learning does not generalize strongly within the same domain." ICLR 2022

---

> ### Author Rebuttal · Authors · 2023-08-09
>
> We thank reviewer 5Ysh for reading our paper, for commending the novelty and relevance of our approach, and for asking intriguing questions. We have clarified some concerns in the global response to all reviewers. Nonetheless, we would like to address some specific points below:
>
> 1. We concur with reviewer 5Ysh that “attribute compositions” are a very interesting and relevant setting. We hope to have illustrated in the **additional examples for possible composition functions in points 1-4 of the global response** and with our **added experiment** that **this is indeed captured by our setting** and that **the compositional data generation does not require multiple objects**. We selected the multi-object (multi-sprite) setting due to its growing significance in ML research and its suitability for visualizing the various conditions to the reader. That said, we emphasize again that we assume a *continuous* data domain with differentiable mappings, which is not applicable to most NLP tasks.
> 2. (Question 1) **Fixed citation in line 23**.
> 3. (Question 2) We **revised Figure 1** and included it in the pdf of the global rebuttal; we hope it is much clearer now.
> 4. (Question 3) **Fixed typo in line 100**.
> 5. (Question 4) In our multi-sprite running example, $z_1^0$ and $z_2^0$ referred to the base configuration of the components in the orthogonal sampling scheme. We indeed failed to introduce this notation properly and **have since clarified this in the text**, thanks for catching it!
> 6. (Question 5) We agree that the relationship between the figures is slightly confusing. Figure 3 illustrates the concept of *compositional support* for 2 components with 1-dimensional marginals (e.g. 2 components with a single attribute). The same concept applies to the 2 components with 5-dimensional marginals (5 attributes shape, color, x, y, …) although this is of course not easily visualized. Figure 4 instead visualizes a 2d projection of the 10-dimensional latent space in order to pinpoint the example images. In the projection, the color, shape, x-position and scale dimensions are not shown explicitly and the axes correspond to the sprites’ y-position. **We have updated the figure captions to clarify the difference.**
> 7. (Limitation 1) **We have added a schematic diagram for the monolithic and compositional models in the appendix** which we omit from the pdf response due to space constraints. As outlined in the experiments section, the compositional model uses the same architecture for each $\varphi_k$: 4 fully-connected layers with 256 hidden units map the latent to a 32×4×4 feature map which is followed by 4 Upsample+Convolution layers with 32 channels mapping to a 3×64×64 slot outputs. The slot outputs are combined using the composition $C$. The monolithic model uses the same number of layers but with 320 hidden units and 64 channels to match the number of parameters in the compositional model.
> 8. (Limitation 2) We thank the reviewer for pointing out interesting work on compositional learning. The architectures presented in [1, 2] are LSTM based and are used for autoregressive, discrete sequence-to-sequence modelling. On the other hand, VAE-based architectures in [1, 3] focus on unsupervised representation learning. Our work instead focuses on the supervised regression setting in a continuous data domain. As outlined in the global response in point 5, a translation to (discrete) NLP tasks or unsupervised settings is not straight forward. However, **it is worth noting that the models employed in our experiments incorporate convolutional and fully-connected layers, mirroring the architecture of VAE decoders as used in [1, 3] and analogous to the regression models tested in [3]**.
> 9. (Limitation 2) The reviewer is correct that [3] is highly correlated with our setting. In fact, **[3] was a direct inspiration for this work as stated in the introduction and the first part of the related work**. The failure (and difficulty) of compositional generalization (even in regression settings) that our work is addressing was first demonstrated in [3]. However, [3] did not provide a thorough analysis of why generalization fails or provided conditions under which it is possible. We believe our work to be the first to address this in a comprehensive manner.
>
> [1] Xu, Zhenlin, Marc Niethammer, and Colin A. Raffel. "Compositional generalization in unsupervised compositional representation learning: A study on disentanglement and emergent language." *NeurIPS 2022*
>
> [2] Ren, Yi, et al. "Compositional languages emerge in a neural iterated learning model." ICLR 2020
>
> [3] Schott, Lukas, et al. "Visual representation learning does not generalize strongly within the same domain." ICLR 2022

---

> > ### Comment · Reviewer_5Ysh · 2023-08-13
> > **Thanks for the response, which addresses most of my concerns.**
> >
> > Thanks very much for the authors' efforts in the rebuttal phase. I still believe the experiments part of the paper needs to improve to make the results more persuasive. I agree that the analysis is based on the continuous assumption of and hence some NLP dataset is not suitable for the evaluation, but what about 3dShapes and MPI3D, which are also commonly used in the field of disentanglement? However, as most of my concerns are addressed, and also considering the theoretical part of the paper, I would increase my evaluation from 5 to 6.

---

> > > ### Author Response · Authors · 2023-08-14
> > >
> > > We thank the reviewer for their reply and for adjusting the score.
> > >
> > > We understand that additional experiments are always desirable. Given the **“novelty of the theoretical results from the paper”** (yWdt), the reviewers’ consensus that **“the paper studies a very important problem from a unique perspective”** (5Ysh), the **“general form”** and **“wide application”** of the theory (1tYF), as well as the impression that the experimental section is **“solid”** and **“provides compelling evidence”** (zapL) we believe these experiments to not greatly strengthen our contribution. Additionally, we highlight the following issues with the proposed experiments:
> > >
> > > - Both MPI-3D and Shapes3D are pre-generated datasets; we do not have access to the data generator and, therefore, cannot sample at will as in our experimental setup. Our theorem assumes that the Jacobian $\frac{\boldsymbol C}{\boldsymbol z}$ can be estimated everywhere on the training distribution, which implies:
> > >     1. The **latents should be sampled from a continuous range**. However, Shapes3D has 6 latents which can only assume between 4 and 15 discrete values; MPI3D’s latents are similar. While our training setting also includes *one* discrete factor (shape with 3 possible values), indicating that it is not impossible to train a model in this case; in contrast, *all* factors in the proposed datasets are discrete.
> > >     2. The **latents should be sampled densely**. In the orthogonal sampling case with 2 components, we train with 50,000 samples per component for a total of 100,000 data points. In Shapes3D, assuming we treat each latent independently, the model would have to be fitted on only 4 to 15 samples per factor for a total of 57 data points. Even if we clubbed multiple latents together, e.g., treated `object_hue`, `scale`, `shape`, and `orientation` as a single “object” component, this would lead to only 4,800 samples for this component, still 10× less than in our setting. We arrive at similar numbers for MPI3D.
> > >
> > >     Taken together, this makes fitting a model on these datasets difficult in practice, and we expect low reconstruction quality, even ID. In this case, **we cannot draw clear conclusions about the model’s generalization capability**.
> > >
> > > - To model the composition of latent factors in 3D space, $\boldsymbol C$ would need to be implemented as a much more complex function. An example of this is illustrated in Fig. 3 of [1], which highlights the complexity of the resulting composition, including ray-casting, 3D transformations, and camera projections. Besides, even though [1] deals with a slightly different training setting, the authors show that training with the inductive bias of such a composition function does indeed enable some generalization (compare Fig. 8). Therefore, **we believe implementing such a model ourselves would be a significant overhead for little added value**.
> > >
> > > If the reviewer maintains that the experiments would nonetheless greatly increase the value of our work, we will look into implementing them for the final version of the paper.
> > >
> > > [1] https://arxiv.org/abs/2011.12100

---

### Official Review · Reviewer_SW1j · 2023-07-05

**Soundness:** 2 fair
**Presentation:** 2 fair
**Contribution:** 2 fair
**Rating:** 5
**Confidence:** 3

**Summary:**

This paper proposes formal conditions on a data generating procedure such that "compositional generalization", i.e. generalization to unseen combinations of elements observed during training, can be shown to be possible. The authors construct a toy task and show empirically that when the proposed conditions are met, models can generalize in the predicted way.

**Strengths:**

* The term "compositional generalization" is often only intuitively defined in prior work. A better theoretical understanding of the constraints between training and test distributions would be helpful. This paper proposes a theoretical framework regarding what conditions must be met regarding the training and test distributions for compositional generalization to be possible, albeit for a relatively limited case and for an implicitly considered class of models.

**Weaknesses:**

* The case considered is a very limited form of compositional generalization, motivated by decomposing images into multiple objects. This is fine, but I think the limited scope of the paper and theory should be clearer, unlike in the current paper, which includes claims such as "conditions... which are sufficient for compositional generalization". It does not seem like the proposed model would be a good fit for the data generating procedure of more commonly studied cases of compositional generalization, such as those arising in mapping natural language to its semantics (e.g. https://arxiv.org/abs/1711.00350). In these case, the data generating procedure has a more hierarchical, potentially recursive, compositional structure.
* The empirical experiments are limited to a toy dataset constructed by the authors. It would be more compelling if the authors could show how the proposed conditions apply to existing datasets that have been proposed (and accepted by the community) as evaluating "compositional generalization".
* The proposed conditions seem rather strong. The limitations section mentions that models with various inductive biases could allow relaxation of some of the proposed conditions. At the same time, the authors implicitly assume some strong conditions on the model, e.g. that it factors similarly to the proposed model for the data generating procedure. A more satisfying theoretical framework might more explicitly consider what inductive biases are necessary for the model to be sufficiently specified by the training data such that it generalizes to the test distribution.

Nits:

* Line 23 - missing reference
* The notation used in equation 2 was not clear to me. What is $z_k^0$? I didn't see where the superscript notation $^0$ for the latents was defined.
* The notation used in equation 3 was not clear to me. The `supp` operator defined in equation 2 does not have a subscript, but equation 3 introduces a subscripted latent.
* Line 149 - distribution -> distributions

**Questions:**

See weaknesses above.

**Limitations:**

Yes

---

> ### Author Rebuttal · Authors · 2023-08-09
>
> We thank reviewer SW1j for reviewing our paper and for lauding our effort to formalize notions of compositional generalization. While we hope to have already captured most of your concerns in our general reply above, we would like to also address your individual concerns here.
>
> 1. (Weakness 1): Our work primarily addresses compositional generalization on the continuous supervised regression setting. We concur with the reviewer's perspective that there's room for improving the emphasis on this aspect, and **have adapted the introduction and method sections as well as the discussion to more clearly explain the limitations of our work**. We have included these limitations in the global reply points 5-7. As explained there, our framework does not readily extend to NLP data generating processes like the one in [1]. We maintain, however, that our theory is applicable to a wide range of compositions in various data domains and tasks, as outlined in the global reply in points 1-4.
> 2. **We have conducted an additional experiment** with different component and composition functions to further demonstrate the generality of our theorem. For more details we kindly direct the reviewer's attention to points 8 and 9 in the global response and the accompanying rebuttal document. We acknowledge that our new experiment remains within the image domain but maintain the conviction that our claims hold true across various domains. As highlighted in the concluding section of our global response, we kindly ask to judge the value of our contribution based on the theoretical results which make up the bulk of our work and greatly improve upon prior works. The application of our results to more complicated real-world datasets which are substantially harder to analyze is a direction we intend to explore in subsequent works.
> 3. We thank the reviewer for pointing out the interesting work [2]. We acknowledge that we are not the first to give a formal definition for compositional generalization. However, as also pointed out by the reviewer, such a formalization requires tractable assumption to derive any theoretical guarantees, which the formulation in [2] is lacking. It's crucial to emphasize that our study's objective is to establish a formal framework for addressing compositional generalization within the context of regression to continuous data domains. **We kindly request the reviewer to elaborate how a formalization as in [2] could facilitate this**. We rather see our work as a continuation of [3, 4], which we cite in our introduction and related works section. Compared to these earlier works, our method allows for much more a more versatile function class with minimal assumptions on the training distribution.
> 4. As addressed in our reply to point 3. and outlined in points 1-4. and the final paragraph of our global response, we maintain that our proposed conditions are much weaker than those of previous works [3, 4]. We acknowledge that we trade-off some assumptions on the data distribution for slightly stronger conditions on the data-generating process, but we do not agree with the reviewer that one kind of assumption is “more theoretically satisfying”. In practice, conditions on the model are much easier to enforce and are often key in identifiability settings which our work aims to lay the foundation for. **We kindly request the reviewer to elaborate why conditions on the data distribution should be preferrable to conditions on the function class.**
>
> [1] https://arxiv.org/abs/1711.00350
>
> [2] https://arxiv.org/abs/2112.07610
>
> [3] https://arxiv.org/abs/2211.11719
>
> [4] https://arxiv.org/abs/2304.14329

---

> > ### Comment · Reviewer_SW1j · 2023-08-16
> >
> > Thank you for your response. I believe the proposed changes will improve the paper and clarify its scope and limitations.
> >
> > > We kindly request the reviewer to elaborate how a formalization as in [2] could facilitate [theoretical guarantees].
> >
> > As I stated in my review, "Unlike such prior work, the authors derive testable formal constraints". My original comment was just intended to add context to the review and connect this work to some literature I was familiar with. I will edit my review to reflect that this comment does not reflect a weakness.
> >
> > > "We kindly request the reviewer to elaborate why conditions on the data distribution should be preferrable to conditions on the function class."
> >
> > I don't think I claimed this. My main point was that "the proposed conditions seem rather strong", as they relate to both the data generating procedure, the model class being considered, and the supervision required.
> >
> > > Other datasets
> >
> > I see some discussion with Reviewer 5Ysh on this topic as well. Even if it is not feasible show how the theorem relates to other real world datasets, it would be useful to at least show how the formalism relates to previously proposed synthetic datasets where the data generating procedure is known.

---

> > > ### Author Response · Authors · 2023-08-16
> > >
> > > We thank the reviewer for the clarification of point 3 and for adapting their review in this regard.
> > >
> > > We also accept the clarification of point 4 and agree that we have misunderstood the reviewer’s original point. With regards to “the proposed conditions seem rather strong” we highlight that
> > >
> > > - the stronger conditions on the data generation process allow for very mild conditions on the data distribution
> > > - conditions on model class are easier to enforce in practice than conditions on the distribution
> > > - compared to prior work, our function class and data generation are much more widely applicable (e.g. [1] only allowed for additive compositions), which other reviewers seem to agree with (e.g., 1tYF: “The theorem has a general form, indicating potential wide application”)
> > >
> > > With respect to experiments on other datasets, besides the additional experiment provided in the rebuttal pdf, we also refer the reviewer to the discussion thread with reviewer yWdt, where we discussed possible options and agreed to look into a CLEVR-like setting for the camera-ready version.
> > >
> > > In light of this, we hope that you can agree that the theoretical result that constitutes our main contribution is sufficiently well validated to be of interest to the community and form a foundation for future work; we, therefore, hope you can confidently increase your score and agree to accept our submission.
> > >
> > > [1] [arxiv.org/abs/2211.11719](http://arxiv.org/abs/2211.11719)

---

> > > > ### Comment · Reviewer_SW1j · 2023-08-18
> > > >
> > > > Thanks for the response. I think showing how the formalism could be applied to a CLEVR-like setting would be very useful. I also understand there is not time to complete this during the rebuttal period. However, it is also difficult to weigh future work that may or may not be successful. But I will increase my score from 4 to 5.

---

### Official Review · Reviewer_zapL · 2023-07-07

**Soundness:** 3 good
**Presentation:** 2 fair
**Contribution:** 3 good
**Rating:** 7
**Confidence:** 2

**Summary:**

In this work, compositionality is examined as a fundamental attribute of the data-generating process, rather than being solely associated with the data. By establishing a precise mathematical framework, it aims to comprehensively analyze and understand compositional generalization. Notably, the work identifies a series of sufficient conditions that enable compositional generalization, highlighting the presence of these conditions in various existing approaches. It validates these observations on a synthetic task on images containing sprites with variable appearance.

**Strengths:**

- This work delves into the exploration of compositionality as a characteristic inherent to the data generating process, drawing inspiration from the literature on identifiable representation learning. To the best of the reviewer's knowledge, this is the first work that approaches compositionality from this particular perspective.

- The synthetic experimental setup, despite its limited scope, is solid; and the results are convincing with reported mean and standard deviation over multiple seeds.

- The ablation study conducted in lines 238-247 provides compelling evidence that violating specific conditions leads to a noticeable decline in compositional generalization. This observation is further supported by Figure 5, which clearly depicts instances of generalization failure precisely where the conditions are violated. These combined findings reinforce the validity of the conclusions.

- Further details are provided in the supplementary material, and anonymized code facilitates the reproducibility of the experiments.

**Weaknesses:**


- One weakness of this work is that all the conducted experiments focus on a single task. While the findings are valuable and provide insights, it would be beneficial to validate them on a  wider range of tasks. By demonstrating the generalizability of the observations across multiple tasks, and potentially across domains beyond images, the overall strength and impact would be significantly enhanced.

- Table 1 presents the performance metrics of both the in-domain (ID) test set and the complete test set (all), which comprises both the ID and the out-domain (OD) test set. To enhance readability and hihglight the individual contributions of ID and OD to the overall performance, it would be beneficial to include separate columns specifically dedicated to the OD performance.

Minor:
- Broken reference at line 23


**Questions:**

- Can this approach provide any insights into compositional generalization in scenarios where the ground-truth data generating process $\mathbf{f}$ operates on latent factors $\mathcal{Z}$ that are either unknown or only partially known (e.g. only the number of factors is known)? Providing more insight and possible future directions in lines 259-264 would be appreaciated.

- Is it possible that there exists a threshold in model complexity or train dataset size beyond which compositional generalization naturally emerges?

- In line 161, what is the meaning of the composition function $\mathbf{Id}$? Is is assumed to be the sum?

**Limitations:**

The limitations are adequately discussed in Section 5 and 6.

---

> ### Author Rebuttal · Authors · 2023-08-09
>
> We express our gratitude to reviewer zapL for their thorough review of our paper, for highlighting the paper’s strengths, and for providing valuable suggestions. While we have already tackled a number of concerns outlined in our global response and will proceed to address specific questions below.
>
> 1. (Weakness 1) **We have conducted an additional experiment** with different component and composition functions to further demonstrate the generality of our theorem. For more details we kindly direct the reviewer's attention to points 8 and 9 in the global response and the accompanying rebuttal document. We acknowledge that our new experiment remains within the image domain but maintain the conviction that our claims hold true across various domains. We kindly refer the reviewer to points 1-4 in the global response, where we **illustrate how our theory can be applied across multiple tasks and domains**.
>
> 2. (Weakness 2): We understand that it is desirable to see the “pure” OOD performance in addition to the performance on the entire test set. However, as the training sets in most cases are planes through a high-dimensional latent space, the **ID samples in the entire test set generally make up less than 0.1% of samples**, so that the “pure OOD” performance would does not differ significantly from the “all” performance. The suggestion to make the table more readable was valuable however, and **we now report the ID and all performance, as well as the gap between them. We’ve also reformatted the standard deviations to be less obtrusive and relegated the MSE to a second table in the appendix**. We’re happy to include the reformatted table in the camera ready version, but couldn’t fit it in the rebuttal pdf due to space constraints.
>
> 3. (Minor Weakness) **We fixed the reference in line 23**.
>
> 4. (Question 1) As explained in point 7 of our global response extension to an unsupervised setting is far from trivial.
>
> 5. (Question 2) This is an interesting question. The theoretical setting doesn’t make assumptions on the sample density in the training distribution, in principle allowing for infinite samples. Even then, we only provide *sufficient* conditions, which means we cannot rule out out the possibility that compositional generalization might always emerge for large, complex models or for models with certain architectures. Outside of the infinite-sample setting, we cannot make any claims about the relationship between training set size and generalization capability. This relationship would most likely depend on the specific densities of the training distribution, on which we deliberately don’t pose any assumptions. We hope to extend our work to such settings in the future.
>
> 6. (Question 3) **Fixed line 161**, $Id$ should be $\sum$; thanks for catching that!

---

> > ### Comment · Reviewer_zapL · 2023-08-13
> >
> > Thanks for the rebuttal! It addresses the comments and questions raised in the review. Therefore, I confirm my score.

---

### Official Review · Reviewer_yWdt · 2023-07-26

**Soundness:** 3 good
**Presentation:** 2 fair
**Contribution:** 3 good
**Rating:** 6
**Confidence:** 4

**Summary:**

This work combines the compositional generalization problem with the data generation process and mathematically defines and formalizes the relevant framework. In this process, two main conditions are proposed: the first is the condition for a given problem to be considered a compositional generalization problem (compositional support), and the second is the condition that ensures the compositional generalization problem can be compositionally generalized (sufficient support). In the proposed theorem, the authors proved that the approximated composition function generalizes to unseen data if two main support conditions are met. In the experiments, the proposed theories are validated by training models under various train set distributions and evaluating their performance on out-of-distribution test sets that include novel compositions.

**Strengths:**

1. The concepts that have been implicitly used in the field of Compositional Generalization are mathematically defined and formalized, which can be valuable for future work related to compositional generalization.
2. The theorem related to sufficient support is quite interesting.

**Weaknesses:**

1. The attempt to support and validate the previously stated theories through experiments is commendable, but the connection seems somewhat weak. For example, while Theorem 1 describes the sufficient condition in a concrete manner using mathematical tools, the concepts of broad support and narrow support used in the experiments appear to be somewhat qualitative. It would be interesting to know if there were specific mathematical criteria or other considerations involved in determining the broadness of the support.
2. It would be beneficial to have more explanation about the Collapsed Composition Function in the Violation Condition section. The provided explanation alone does not entirely justify why RGBa performs better. Specifically, in the case of RGBa, the performance seems not only to be comparable to other cases that satisfy different sufficient conditions but also noticeably better. If it can still be understood that RGBa achieves compositional generalization even with similar performance, it becomes acceptable for the performance to be somewhat lower. However, the part where performance is significantly better remains questionable, and it would be interesting to know if there are additional hypotheses to explain this phenomenon.
3. More experiments are required for empirical validation of the theorem. There are some datasets that consists of diverse factors for compositional generation (e.g., MPI3D, Shapes3D). It would be nice to see the results on those more complex and diverse-factored scenes. Also, there can be various types of composition functions that include factors (e.g., noise, illumination) in real nature. The experimental results on how the sufficient support can be satisfied on different composition functions can be done.
4. The theorem assumes the ground truth latent and the known composition function. While the support condition in unsupervised setting is harder to derive, empirical experiments can be done using scene factorization methods ([1]) and various composition functions.

**Questions:**

1. Are there any other examples that explain the relationship between composition function and sufficient support such as occlusion?

**Limitations:**

Authors mentioned limitations in the paper, while more empirical experiments for validating the theorem seem necessary.

[1] Neural Systematic Binder (ICLR 23)

---

> ### Author Rebuttal · Authors · 2023-08-09
>
> We thank reviewer yWdt for reviewing our paper and providing helpful suggestions. While we hope to have already captured most of your concerns in our global reply above, we would like to also address your individual concerns here.
>
> 1. (Weakness 1.) We agree that the explanation of *broad* and *narrow* supports in the experiment was purely qualitative and should have been more detailed and rigorous. **We now include the specific mathematical criteria and considerations involved in determining the broadness of the support** in a **newly added Appendix C “Details on sufficient support assumption”**. The corresponding sections now read:
>
>     **Orthogonal sampling** As explained in the main text and illustrated in Figure 4, occlusions make the output image invariant to changes of the background sprite in the occluded pixels. The model does not receive a training signal for these pixels and reconstruction becomes impossible. In the case of orthogonal sampling, the foreground sprite is fixed in its base configuration (orange triangle in the bottom left corner) whenever the background sprite is sampled randomly, see first row of the **newly added Figure 6**. The foreground sprite therefore always occludes the same background pixels, for which the Jacobian $\frac{\partial\boldsymbol C}{\partial\boldsymbol \varphi_{km}}$ is zero. Therefore, for each configuration of the background sprite a second data point is required in which the foreground sprite occludes a different and distinct set of pixels (cyan square in the second row of Figure 6). We can clearly observe that this is the case here; therefore, the Jacobians $\frac{\partial\boldsymbol C}{\partial\boldsymbol \varphi_{km}}$ from both samples have zero entries at different indices $m$ and their sum has full rank. The resulting sufficient support has the shape illustrated on the left of Figure 5.
>
>     **Diagonal sampling** As explained above, sufficient support for occluding sprites is guaranteed if the dataset contains at least two samples with the same background sprite and different foreground sprites that overlap distinct pixels (as also illustrated in Figure 6). In the diagonal sampling case we therefore need to choose the width of the diagonal broad enough to guarantee that it contains such foreground sprites. The smallest possible width of the diagonal can be determined by finding the smallest possible x-offset (or y-offset) for which a pair of sprites of the smallest scale does not overlap. This is the case for a width of 0.2, which was used for the *diagonal* sampling case in Table 1 #6. The *broad diagonal* sampling case Table 1 #8 used a width of 0.4 (double the minimal width), while the *narrow diagonal* sampling case in Table 1#10 used a width of 0.1 (half the minimal width).
>
>
> In the same appendix, **we now elucidate the relationship between composition function and sufficient support** (Question 1) for other examples and data modalities, as also briefly outlined in the global response.
>
> 2. (Weakness 2.) We agree with the reviewer that “RGBa [performs] noticeably better”, even OOD. The example was included to illustrate that there exist models which do not satisfy all the conditions of our theorem while still generalizing decently. This is not surprising, as our theorem merely states *sufficient conditions* which enable generalization, not *necessary conditions* that prohibit generalization if violated. We believe that the existence of such an example does not diminish the usefulness of having conditions that guarantee generalization. We included this example in the hope that future work can build on our results derive weaker sufficient conditions. **We acknowledge that this part was confusingly worded in the paper and adjusted the section accordingly**. As to why RGBa performs slightly better: We believe that the parameterization of the problem as alpha compositing (as detailed in Equation 26 in Appendix B) is slightly more amenable to training with SGD since no approximation of step functions (see Equations 25 and 26 in Appendix B) is necessary. In the finite-sample-finite-width training setting, RGBa is then able to fit the data slightly better. However, **the gap in performance is practically identical to the RGB case (0.005 vs. 0.006)**, indicating that the model does not “generalize better”.
> 3. (Weakness 3 and 4) **We have conducted an additional experiment** with different component and composition functions to further demonstrate the generality of our theorem. For more details we kindly direct the reviewer's attention to points 8 and 9 in the global response and the accompanying rebuttal document. As highlighted in the concluding section of our global response, we kindly ask to judge the value of our contribution based on the theoretical results which make up the bulk of our work and greatly improve upon prior works. The application of our results to more complicated real-world datasets which are substantially harder to analyze is a direction we intend to explore in subsequent works.
> 4. (Weakness 4.) As explained in point 6 of our global response extension to an unsupervised setting is far from trivial.

---

> > ### Comment · Reviewer_yWdt · 2023-08-13
> >
> > Thanks for answering to the reviews in detail!
> >
> > (Weakness 1.) The detailed explanations on broadness of the support seem reasonable.
> >
> > (Weakness 2.) As the authors mentioned, please adjust the section from the response.
> >
> > (Weakness 3.) It's nice to see additional experiments on different composition functions. The result looks acceptable. Although the reviewer acknowledges the novelty of the theoretical results from the paper, it would be more powerful to see the verification of the theorem in various dataset (as mentioned Shapes3D or MPI3D which are not real-world scale dataset). I would be pleased to see the similar results on other dataset.
> >
> > (Weakness 4.) I agree. It looks like this is out of the scope.

---

> > > ### Author Response · Authors · 2023-08-14
> > >
> > > We thank the reviewer for their reply and for commending our added experiment.
> > >
> > > (Weakness 3) We understand that additional experiments are always desirable. Given the **“novelty of the theoretical results from the paper”** (yWdt), the reviewers’ consensus that **“the paper studies a very important problem from a unique perspective”** (5Ysh), the **“general form”** and **“wide application”** of the theory (1tYF), as well as the impression that the experimental section is **“solid”** and **“provides compelling evidence”** (zapL) we believe these experiments to not greatly strengthen our contribution. Additionally, we highlight the following issues with the proposed experiments:
> > >
> > > - Both MPI-3D and Shapes3D are pre-generated datasets; we do not have access to the data generator and, therefore, cannot sample at will as in our experimental setup. Our theorem assumes that the Jacobian $\frac{\boldsymbol C}{\boldsymbol z}$ can be estimated everywhere on the training distribution, which implies:
> > >     1. The **latents should be sampled from a continuous range**. However, Shapes3D has 6 latents which can only assume between 4 and 15 discrete values; MPI3D’s latents are similar. While our training setting also includes *one* discrete factor (shape with 3 possible values), indicating that it is not impossible to train a model in this case; in contrast, *all* factors in the proposed datasets are discrete.
> > >     2. The **latents should be sampled densely**. In the orthogonal sampling case with 2 components, we train with 50,000 samples per component for a total of 100,000 data points. In Shapes3D, assuming we treat each latent independently, the model would have to be fitted on only 4 to 15 samples per factor for a total of 57 data points. Even if we clubbed multiple latents together, e.g., treated `object_hue`, `scale`, `shape`, and `orientation` as a single “object” component, this would lead to only 4,800 samples for this component, still 10× less than in our setting. We arrive at similar numbers for MPI3D.
> > >
> > >     Taken together, this makes fitting a model on these datasets difficult in practice, and we expect low reconstruction quality, even ID. In this case, **we cannot draw clear conclusions about the model’s generalization capability**.
> > >
> > > - To model the composition of latent factors in 3D space, $\boldsymbol C$ would need to be implemented as a much more complex function. An example of this is illustrated in Fig. 3 of [1], which highlights the complexity of the resulting composition, including ray-casting, 3D transformations, and camera projections. Besides, even though [1] deals with a slightly different training setting, the authors show that training with the inductive bias of such a composition function does indeed enable some generalization (compare Fig. 8). Therefore, **we believe implementing such a model ourselves would be a significant overhead for little added value**.
> > >
> > > If the reviewer maintains that the experiments would nonetheless greatly increase the value of our work, we will look into implementing them for the final version of the paper.
> > >
> > > [1] https://arxiv.org/abs/2011.12100

---

> > > > ### Comment · Reviewer_yWdt · 2023-08-15
> > > >
> > > > I agree that there is no data generating code for Shape3D, and MPI3D.
> > > >
> > > > I found that data generating code for the CLEVR dataset [1] is available.
> > > >
> > > > While authors noted that empirically testing the theorem on complex dataset (such as CLEVR) would be done in future work, it would be nice to see the results on this with simple setting in manuscript (may cover occlusion case only) in the final version.
> > > >
> > > > [1] https://github.com/facebookresearch/clevr-dataset-gen

---

> > > > > ### Author Response · Authors · 2023-08-16
> > > > >
> > > > > We thank the reviewer for pointing out the data generation for the CLEVR dataset.
> > > > >
> > > > > We understand that performing an experiment on a 3D dataset can verify the applicability of our theoretical framework in addition to the experiments we already presented. Given the tight timeline of this rebuttal phase, we cannot produce results now but will look into this for the camera-ready version of our paper.
> > > > >
> > > > > Having said that, we would like to point out that we see the main contribution of our work in our theoretical results. We hope that you agree with us that our theoretical results and experimental validations are already sufficiently strong to advance the understanding of the community and hope that you can, therefore, confidently increase the score and recommend accepting our submission.

---

> > > > > > ### Comment · Reviewer_yWdt · 2023-08-16
> > > > > >
> > > > > > Okay! Thanks for detailed discussion. I raise my score from 5 to 6.

---

### Author Rebuttal · Authors · 2023-08-09

We would like to thank all 5 reviewers for their time and valuable feedback!

We appreciate their assessment of our work as **“valuable for future work related to compositional generalization”** (yWdt). The reviewers concur that **“the lack of compositional generalization ability has been a longstanding problem, and it is good to have a theory for it”** (1tYF) and find that **“the paper studies a very important problem from a unique perspective”** (5Ysh) by addressing a setting which **“is often only intuitively defined in prior work [and for which] a better theoretical understanding […] would be helpful”** (SW1j). The reviewers commend the paper as the **“first work that approaches compositionality from this perspective”** (zapL) which **“has the potential to be extended to more general scenarios”** (5Ysh).

While the some reviewers praised the theorem’s **“general form, indicating potential wide application”** (1tYF) and its ability to **“cover a wide range of existing approaches”** (1tYF), some questions about the applicability of our results to other settings remain. We realize this should have been clearer in the paper and provide a short list of examples here that we **have included in a new section.**

The theorem **can cover**, e.g.:

1. **Additive compositions with occlusion** as shown in the paper. This composition function can model many object-centric datasets [2]. Even this simple composition function was outside the scope of prior work [3, 4]. We chose it as running example in the paper for ease of visualization.
2. **Attributes/transformations of an object**, e.g. with components shape, translation, color of a sprite composed in pixel space by a convolution or matrix multiplication. Or the components could be scale, shear, rotation, translation, projection in 3d applied to an object’s mesh by matrix multiplications. *Sufficient support* is violated whenever the parameterized transformations produce invariances, e.g. when a circular object should be rotated and the gradient of the rotation component cannot be reconstructed from the output.
3. The components could be **audio signals** that are overlayed with a $\max$ operation to model finite microphone gain. *Sufficient support* is violated if one signal saturates the microphone and drowns out the other.
4. **Vector or matrix products** The components are arbitrary vectors or matrices and the composition function is their (outer) product. *Sufficient support* is violated if an element (row/column) in the second vector (matrix) is zero. In that case the multiplied element (column/row) in the first vector (matrix) cannot be reconstructed.

The theorem **cannot cover:**

5. **Discrete data domains** such as NLP tasks or symbolic reasoning, as we assume continuous, differential mappings between the component latents and composition.
6. **Unsupervised Settings**. If data-generating process or latent factors are (partially) unknown, the model’s inferred representation can have an arbitrary relationship to the ground-truth latent space. Before generalization can be addressed, this problem of *identifiability* needs to be solved. There exists no prior work addressing identifiability on restricted training sets $P$. An extension of our results to unsupervised settings is, therefore, far from trivial. As prior work [1] suggested, generalization is already an interesting and challenging problem in the regression setting we examine. We are confident that our results constitute a **step towards a comprehensive theory of *generalizable identifiability* that encompasses the unsupervised case** and are motivated to address this problem in future work. **We realize that the discussion section was insufficient in explaining this issue and have updated it.**
7. **Hierarchical compositions**, as we consider only a single composition function.

While the reviewers commended our existing **“solid experimental setup”** (zapL), and mentioned that the **“experiments validate the theoretical results”** (1tYF), there were some calls for additional empirical validation. We thank the reviewers for suggesting a setting where each component models a “different characteristic of ONE object” (5Ysh) which **we now implemented and added to the paper (Experiment 2)**:

8. The data consists of square shapes with random rotation, color, and Gaussian blur, each modelled by one component and combined using a convolution operation as the composition function (see Fig. 7). The setting further highlights the generality of our theoretical setting.
9. Since the reviewers asked for “experimental results on how the sufficient support can be satisfied on different composition functions” (yWdt), we assess performance on two sampling schemes: a diagonal one which satisfies *sufficient support* and an orthogonal one which violates it. As explained in points 2. above and illustrated in Fig. 8, violating *sufficient support* leads to invariances in the training data that the model cannot resolve.

Finally, we wish to position our work thoughtfully within the broader literature landscape. **Our main contribution is a theoretical analysis of compositional OOD generalization and we hope the work can be evaluated as such**. There exist only a couple of studies [3, 4] that directly confront this issue, both of which approach it from a different direction (assuming specific distributions) and impose much stronger limitations on the composition function. **We kindly request the reviewers to assess our work within this context.**

[1] [arxiv.org/abs/2107.08221](http://arxiv.org/abs/2107.08221)

[2] [github.com/deepmind/multi_object_datasets#multi-dsprites](http://github.com/deepmind/multi_object_datasets#multi-dsprites)

[3] [arxiv.org/abs/2211.11719](http://arxiv.org/abs/2211.11719)

[4] [arxiv.org/abs/2304.14329](http://arxiv.org/abs/2304.14329)

---

### Decision · Program_Chairs · 2023-09-21

**Decision:**

Accept (poster)

**Comment:**

The paper provides a theory of compositional generalization by taking a bottom-up approach by focusing on the data-generating process rather than the data itself. Through this lens, it offers mild conditions on the training distribution and model architecture that are necessary for achieving compositional generalization and validates these conditions both theoretically and empirically.

Overall, the reviewers acknowledge the paper's strength in presenting a mathematically rigorous and novel perspective on compositional generalization. The theoretical framework and theorems offered are seen as significant contributions that can potentially have a broad impact on future research. However, reviewers have pointed out certain weaknesses, such as the limited scope of empirical experiments and the focus on a single task, which could be improved to strengthen the paper's arguments. Some of these concerns were not fully addressed in the rebuttal. Nevertheless, the paper’s core contributions are solid and provide a novel way to understand an important problem in machine learning. Therefore, the paper is recommended for acceptance.